# An amplified derepression controller with multisite inhibition and positive feedback

Gorana Drobac☯, Qaiser Waheed☯, Behzad Heidari, Peter Ruoff📧*

Department of Chemistry, Bioscience, and Environmental Engineering, University of Stavanger, Stavanger, Norway

☯ These authors contributed equally to this work.
* peter.ruoff@uis.no

## Abstract

How organisms are able to maintain robust homeostasis has in recent years received increased attention by the use of combined control engineering and kinetic concepts, which led to the discovery of robust controller motifs. While these motifs employ kinetic conditions showing integral feedback and homeostasis for step-wise perturbations, the motifs' performance differ significantly when exposing them to time dependent perturbations. One type of controller motifs which are able to handle exponentially and even hyperbolically growing perturbations are based on derepression. In these controllers the compensatory reaction, which neutralizes the perturbation, is derepressed, i.e. its reaction rate is increased by the decrease of an inhibitor acting on the compensatory flux. While controllers in this category can deal well with different time-dependent perturbations they have the disadvantage that they break down once the concentration of the regulatory inhibitor becomes too low and the compensatory flux has gained its maximum value. We wondered whether it would be possible to bypass this restriction, while still keeping the advantages of derepression kinetics. In this paper we show how the inclusion of multisite inhibition and the presence of positive feedback loops lead to an amplified controller which is still based on derepression kinetics but without showing the breakdown due to low inhibitor concentrations. By searching for the amplified feedback motif in natural systems, we found it as a part of the plant circadian clock where it is highly interlocked with other feedback loops.

## Introduction

The concept of homeostasis [1], defined by Cannon in 1929 [2], is fundamental to our understanding how organisms, including our body, work [3]. According to Cannon homeostasis refers to the automatic, self-regulating processes that keep steady states within certain, but narrow limits, despite internal or environmental perturbations [1–3]. Although (negative) feedback is recognized as a central part in homeostatic regulation [1, 4, 5], it is not the only dynamic component. The 'homeostatic response' may also include features such as anticipatory mechanisms [6], feedforward loops [7], or positive feedbacks [3, 8, 9].

**Data Availability Statement:** All relevant data are within the manuscript and its Supporting information files.

**Funding:** The author(s) received no specific funding for this work.

**Competing interests:** The authors have declared that no competing interests exist.

In control engineering robust regulation of a variable $A$ with set-point $A_{set}$ can be achieved by so-called integral control [10], which is able to (precisely) correct for step-wise perturbations acting on a controlled variable $A$ [10] (Fig 1).

While in engineering integral control began to be applied in the beginning of the twentieth century with the power steering of ships [11], its usage in physiology/biology first appeared once cybernetics [12–14] made the analogies between engineered and biological systems more explicit. Physiological models during this first era showed the dynamical processes, loops, integrated errors, etc. mostly in terms of flow diagrams, transfer functions, as engineered systems are described [15, 16]. By the turn of the century, when the molecular biology behind physiological processes became better understood researchers began to describe the control process in terms of their reaction kinetics, such as in integral rein control [17] (focussing on that physiological controllers come in antagonistic pairs; see also the later analogous notion of inflow/outflow controllers [18]), in the integral feedback formulation of robust bacterial chemotaxis [19, 20], or in the integral control approach of blood calcium homeostasis [21].

It became evident that certain kinetic conditions within a negative feedback loop, such as zero-order kinetics [18–20, 22–24], autocatalysis [25–27], or second-order (bimolecular/antithetic) kinetics [28, 29] can lead to robust adaptation [30, 31] by integral control where an intrinsic integration of the error between set-point and the actual value of the controlled variable is automatically performed. When these feedback motifs were investigated towards time-dependent perturbations, it turned out that controller performances can differ significantly, either due to the structure of the feedback loop or due to the kinetics of how the integral controller is implemented [32, 33].

Feedback structures which have been found to perform well when exposed to different time-dependent perturbations are based on derepression kinetics [32].

Fig 2 shows one of the motifs (motif 2) with derepression kinetics acting as an inflow controller [18]. Reactions are color-coded and relate to the scheme in Fig 1. This motif, unlike those not using derepression, is able to adapt perfectly to exponentially increasing perturbations. Motif 2 can even balance hyperbolically increasing disturbances with doubling times which decrease exponentially [32]. In Fig 2 $k_1$ represents a (time-dependent or constant) perturbation and $k_2$ is the maximum fully derepressed compensatory flux. $E$ is subject to an

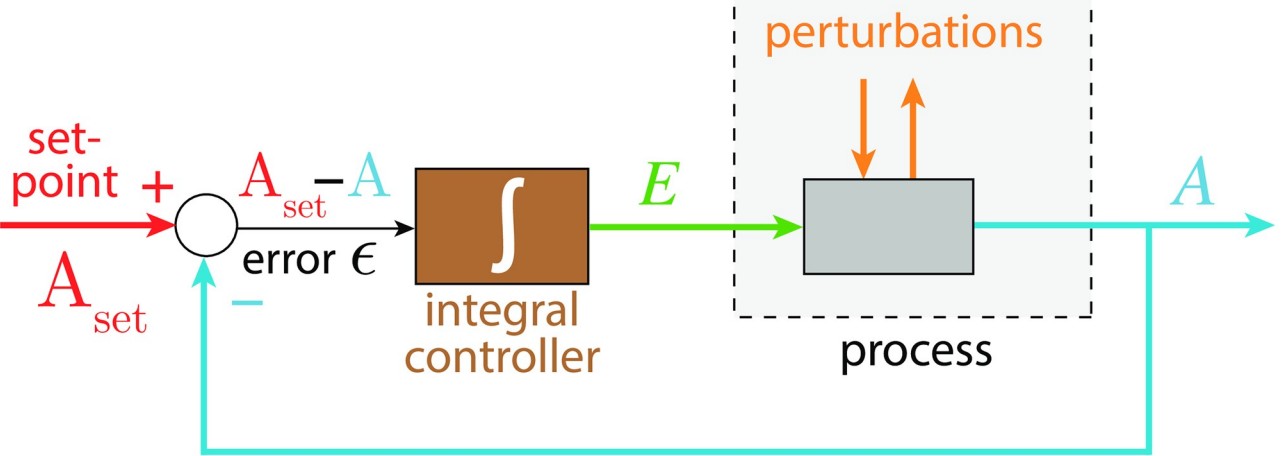

**Fig 1. Scheme of a negative feedback with integral control regulating variable $A$.** The difference between the set-point $A_{set}$ and the actual value of $A$ (the error $\epsilon$) is integrated over time and fed back into the process which generates $A$. This procedure ensures that $A$ will precisely reach $A_{set}$ for stepwise perturbations. Colors correspond to molecular reactions shown later.

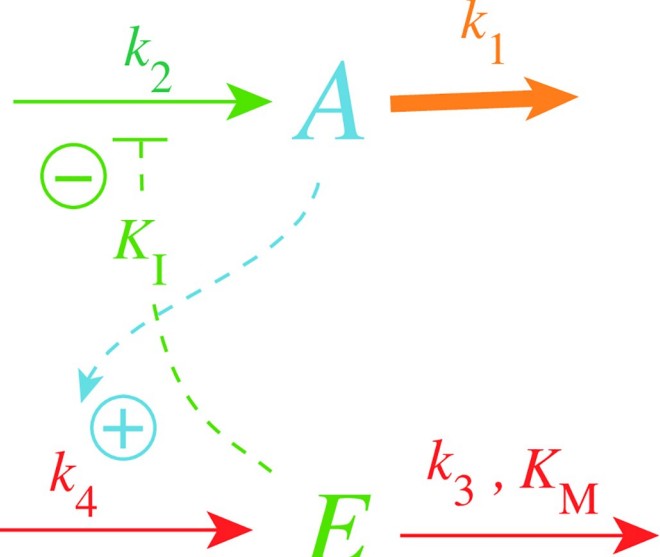

**Fig 2. Inflow controller based on derepression kinetics (motif 2, Ref. [18]).** Red color: reactions determining the set-point; orange: perturbation; blue: *A*-signaling; green: *E*-signaling.

enzymatic zero-order degradation described by the rate parameters $k_3$ ($V_{max}$) and $K_M$, while $k_4$ represents a zero-order synthesis rate with respect to $E$. $K_I$ is an inhibition constant.

The rate equations of the motif 2 controller are:

$$\dot{A} = \frac{k_2}{1 + \frac{E}{K_I}} - k_1 \cdot A \tag{1}$$

$$\dot{E} = k_4 \cdot A - \frac{k_3 \cdot E}{K_M + E} \tag{2}$$

where $k_2/(1 + (E/K_I))$ describes the compensatory flux, which opposes the perturbing flux $k_1 A$. For step-wise changes in $k_1$ and for low $K_M$ values ($K_M \ll E$), $E/(K_M + E) \approx 1$ and the steady state value in $A$ is described by the set-point (setting $\dot{E} = 0$ and solving for $A_{ss}$)

$$A_{ss} = A_{set} = \frac{k_3}{k_4} \tag{3}$$

which will also be defended against time-dependent (increasing) $k_1$ values (see later).

Metaphorically speaking, the activation of the compensatory flux by derepression is somewhat like the static takeoff of an airliner, when the plane stands still at the beginning of the runway, but having engines in full thrust with the breaks on. When the brakes are released the plane starts to accelerate and rapidly reaches takeoff speed.

The controller in Fig 2 reaches its maximum compensatory flux when $E \leq K_I$; any further increase in the perturbation $k_1$ cannot be opposed and will lead to the controller's breakdown. This is illustrated in Fig 3 for different $K_I$ values with $k_1$ increasing exponentially. The $k_1$ value at breakdown, $k_1^{bd}$, can be estimated by setting $E = K_I$ and solving for $k_1$ from Eq 1 with $\dot{A} = 0$, i.e.,

$$k_1^{bd} \approx \frac{k_2}{2A_{set}} \tag{4}$$

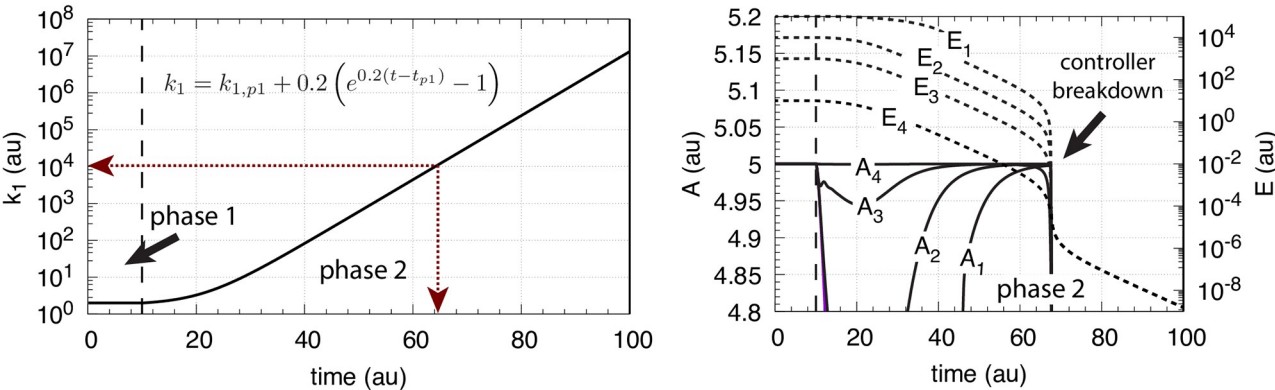

**Fig 3. Behavior of the motif 2 zero-order controller (Fig 2) upon an exponential increase of $k_1$ and the influence of $K_I$.** Left panel shows $k_1$ as a function of time. Phase 1: $k_1 = k_{1,p1} = 2.0$ and constant. The controller is at steady-state at its set-point $A_{set} = 5.0$. Phase 2: $k_1$ starts to increase exponentially at time $t_{p1} = 10.0$ according to the inset Equation. Right panel shows the controller's $A$ and $E$ values for different $K_I$'s, where $A_1$ is $A$ when $K_I = 10.0$, $A_2$ is $A$ when $K_I = 1.0$, $A_3$ is $A$ when $K_I = 0.1$, and $A_4$ is $A$ when $K_I = 1 \times 10^{-3}$. The other rate constants are: $k_2$ (max compensatory rate) = $1 \times 10^5$, $k_3 = 5 \times 10^3$, $k_4 = 1 \times 10^3$, $K_M = 1 \times 10^{-6}$. Initial concentrations: $A_1$-$A_4 = 5.0$, $E_1 = 1 \times 10^5$, $E_2 = 1 \times 10^4$, $E_3 = 1 \times 10^3$, and $E_4 = 10.0$. Controller starts to break down when $k_1$ is reaching $1 \times 10^4$ indicated by the red arrows in the left panel.

The curves $A_i$ and $E_i$ in the right panel show the different $A$ and $E$ values when $K_I$ takes the values 10.0 ($A_1$), 1.0 ($A_2$), 0.1 ($A_3$), and $1 \times 10^{-3}$ ($A_4$). While the controller's "lifetime" (the time span until the controller breaks down) is practically not affected by the different $K_I$ values, the controller's "aggressiveness", i.e. its ability to rapidly respond to perturbations and to keep $A$ at $A_{set}$, is improved with decreasing $K_I$ values.

## Goal of this work

While motif 2 and related controllers based on derepression can keep the controlled variable $A$ at its set-point even for rapidly increasing time-dependent perturbations [32, 33] (Fig 3), they suffer from breakdown once the controller variable $E$ becomes close to or lower than $K_I$. We wondered whether it would be possible to circumvent this restriction to a controller where the control species' concentration increases with increasing perturbation strength, while still keeping the controller properties based on derepression. Using the motif 2 controller as an example, we show that implementation of a positive feedback loop based on autocatalysis combined with multisite inhibition kinetics can avoid controller breakdown by low $E$ values, but still shows the properties of a controller based on derepression.

## Materials and methods

Computations were carried out by using the Fortran subroutine LSODE [34]. Plots were generated with gnuplot (www.gnuplot.info) and slightly edited with Adobe Illustrator (adobe.com). To make notations simpler, concentrations of compounds are denoted by compound names without square brackets. Time derivatives are generally indicated by the 'dot' notation. Concentrations and rate parameter values are given in arbitrary units (au). In the Supporting Information a set of MATLAB (mathworks.com) scripts are provided for illustration in comparison with corresponding Fortran calculations (S1 Matlab).

## Results and discussion

### Effect of multisite inhibition on controller performance

In this section we investigate the effect of multisite inhibition. In mechanistic terms, we consider an enzyme or transporter, which is responsible for the compensatory flux [33]. In

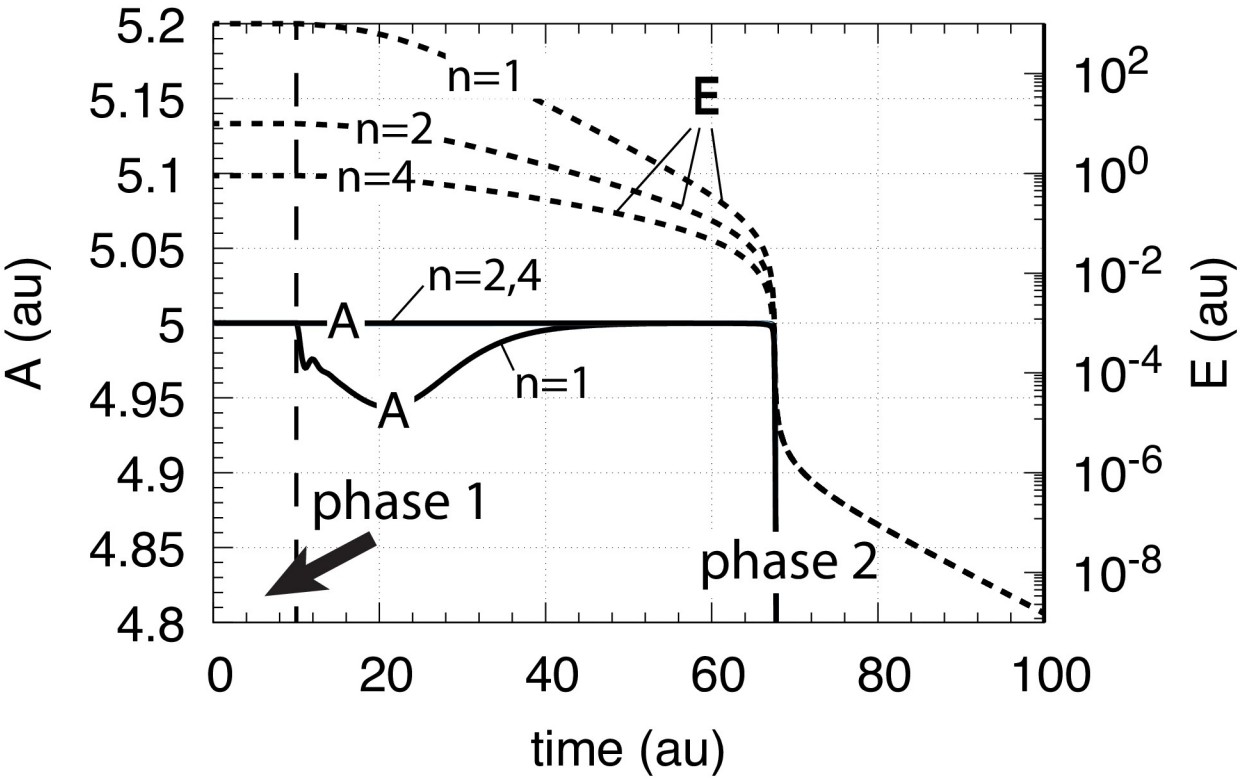

**Fig 4. Behavior of the motif 2 zero-order controller (Fig 2) upon an exponential increase of $k_1$ (left panel Fig 3 and the influence of multisite inhibition (Eq 5)).** Phase 1: the controller is at its set-point/steady state with constant $k_1 = 2.0$. Phase 2: Response of the controller with the different $n$ values upon the exponential increase of $k_1$. Rate constants are: $K_I = 0.1$, $k_2$(max compensatory rate) $= 1 \times 10^5$, $k_3 = 5 \times 10^3$, $k_4 = 1 \times 10^3$, $K_M = 1 \times 10^{-6}$. Initial concentrations: $A = 5.0$ (for all $n$), $E (n = 2) = 9.9$, $E (n = 4) = 0.9$.

multisite inhibition $E$ can bind to the enzyme or transporter at multiple sites with different binding constants $K_I$. To make things more straightforward, we assume that one, two, or four molecules of $E$ can bind to the enzyme/transporter, but always with the same binding constant $K_I$. In this case, Eq 1 is replaced by (see for example Ref [35])

$$\dot{A} = \frac{k_2}{1 + \left(\frac{E}{K_I}\right)^n} - k_1 \cdot A \tag{5}$$

where $n$ is the number of inhibiting $E$ molecules ($n = 1$, 2, or 4). Taking Fig 3 with $K_I = 0.1$ as a starting point, Fig 4 shows the results when $n$ is changed from 1 to 2 and to 4.

From Fig 4 it is clearly seen that multisite inhibition improves the controller's performance, i.e., makes the controller more aggressive by showing a more rapid response and by keeping $A$ closer to the controller's set-point. However, despite the better responsiveness of the controller when using multisite inhibition, the "lifetime" of the controller is not improved, i.e., the break-down occurs at the same time/$k_1^{bd}$ value as in Fig 3.

## Increasing controller lifetime by increasing the maximum compensatory flux

Eq 4 indicates that increasing the maximum compensatory flux $k_2$ will increase $k_1^{bd}$ and thereby increase the lifetime of the controller (upon increasing values of $k_1$). To automate this we have

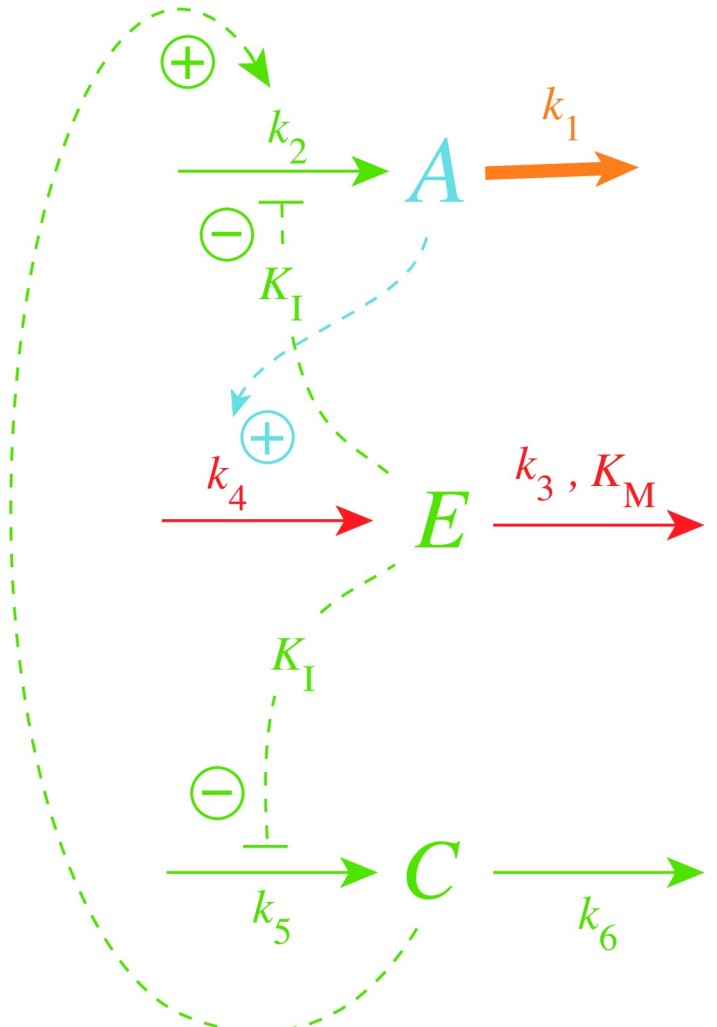

**Fig 5. Increasing controller lifetime by an $E$-dependent increase of the maximum compensatory flux $k_2$.**

added a new variable $C$, shown in Fig 5, which is activated by derepression from $E$, but with the effect to increase $k_2$.

For the sake of simplicity the inhibition constant of the derepression of $C$ by $E$ is assumed to be the same ($K_I$) as for the $E$-induced derepression of the compensatory flux. The rate equations for this controller are

$$\dot{A} = \frac{k_2 C}{1 + \left(\frac{E}{K_I}\right)^n} - k_1 \cdot A \tag{6}$$

$$\dot{C} = \frac{k_5}{1 + \frac{E}{K_I}} - k_6 \cdot C \tag{7}$$

Note that the rate equation for $E$ remains unaltered. We also keep the multisite inhibition of the compensatory flux by $E$, but consider only a single $E$-binding site for the inhibition of the zero-order generation ($k_5$) of $C$. Fig 6 shows the lifetime of controllers with $n = 4$ and with

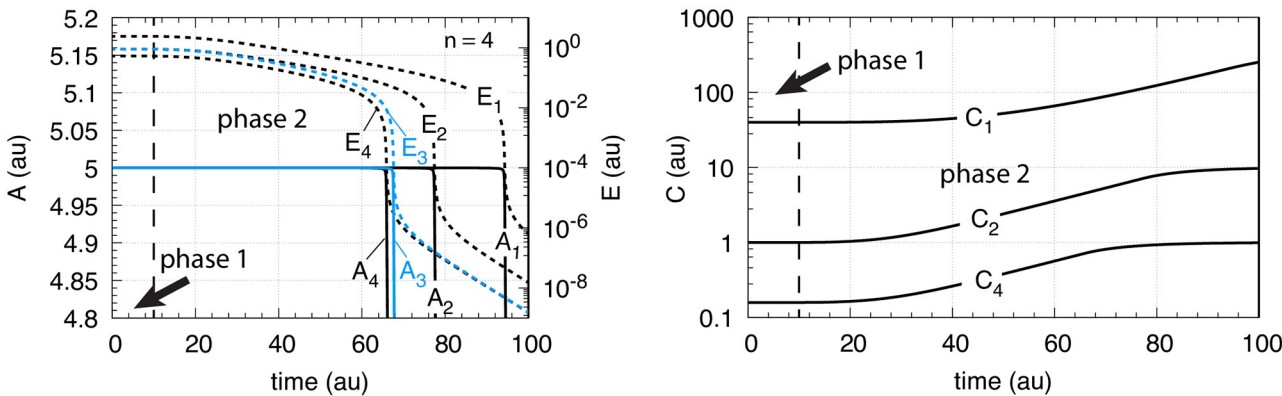

**Fig 6. Activation of the compensatory flux for reaction scheme of Fig 5 and corresponding Eqs 6, 2 and 7 with $n = 4$.** Left panel: Concentrations of $A$ and $E$ as a function of time with different $k_5/k_6$ rate constants. Blue curves $A_3$ and $E_3$ correspond to the calculations without $C$ (Fig 4 with $n = 4$). Right panel: Concentrations of $C$ with different $k_5/k_6$ rate constants as a function of time. Phase 1: the controller is at steady state at its set-point $A_{set} = 5.0$ at constant $k_1 = 2.0$. Phase 2: $k_1$ increases exponentially according to the inset in the left panel of Fig 3. $k_5$ and $k_6$ values and initial concentrations for the different $A_i$, $E_i$, $C_i$ curves: $i = 1$, $k_5 = 10.0$, $k_6 = 0.01$, $A_0 = 5.0$, $E_0 = 2.412$, $C_0 = 39.81$; $i = 2$, $k_5 = 1.0$, $k_6 = 0.1$, $A_0 = 5.0$, $E_0 = 0.9$, $C_0 = 1.0$; $i = 3$, no $C$ (Fig 4 with $n = 4$); $i = 4$, $k_5 = k_6 = 0.1$, $A_0 = 5.0$, $E_0 = 0.531$, $C_0 = 0.158$. Other rate constant values: $k_2 = 1 \times 10^5$, $k_3 = 5 \times 10^3$, $k_4 = 1 \times 10^3$, $K_M = 1 \times 10^{-6}$, $K_I = 0.1$.

different $k_5$ and $k_6$ values when $k_1$ increases exponentially as indicated in the left panel of Fig 3. As a reference, the blue solid and dashed lines show respectively the $A_3$ and $E_3$ values from the results of Fig 4 with $n = 4$, i.e., in the absence of $C$. For high $k_5/k_6$ ratios the lifetime of the controller is clearly increased (see traces $A_1$, $E_1$ and $A_2$, $E_2$), while for a ratio of one ($k_5 = k_6 = 0.1$, traces $A_4$, $E_4$) the lifetime of the controller is slightly reduced.

## Opposing $E$ decrease by positive feedback

While the inclusion of $C$ increases the compensatory flux $k_2 C/(1 + (E/K_I)^n)$ and leads to an improvement in the controller's lifetime, the controller still suffers from the general limitation that once $E$ is driven down to values approaching $K_I$, i.e., the controller breaks down when (setting $E = K_I$)

$$k_1^{\text{bd}} \approx \frac{k_2 C}{2 A_{set}} \qquad (8)$$

(Fig 6). We found, that this trend can be circumvented by including a positive feedback in the generation of $C$ without loosing the dynamic properties of the motif 2 controller. The positive feedback in $C$ can be generated by first-order or second-order autocatalysis.

Fig 7 shows the scheme for both first-order and second-order autocatalysis.

In case of first-order autocatalysis Eq 7 is changed to

$$\dot{C} = \frac{k_5 C}{1 + \frac{E}{K_I}} - k_6 \cdot C \qquad (9)$$

while for the second-order case both the synthesis and degradation terms are second-order with respect to $C$

$$\dot{C} = \frac{k_5 C^2}{1 + \frac{E}{K_I}} - k_6 \cdot C^2 \qquad (10)$$

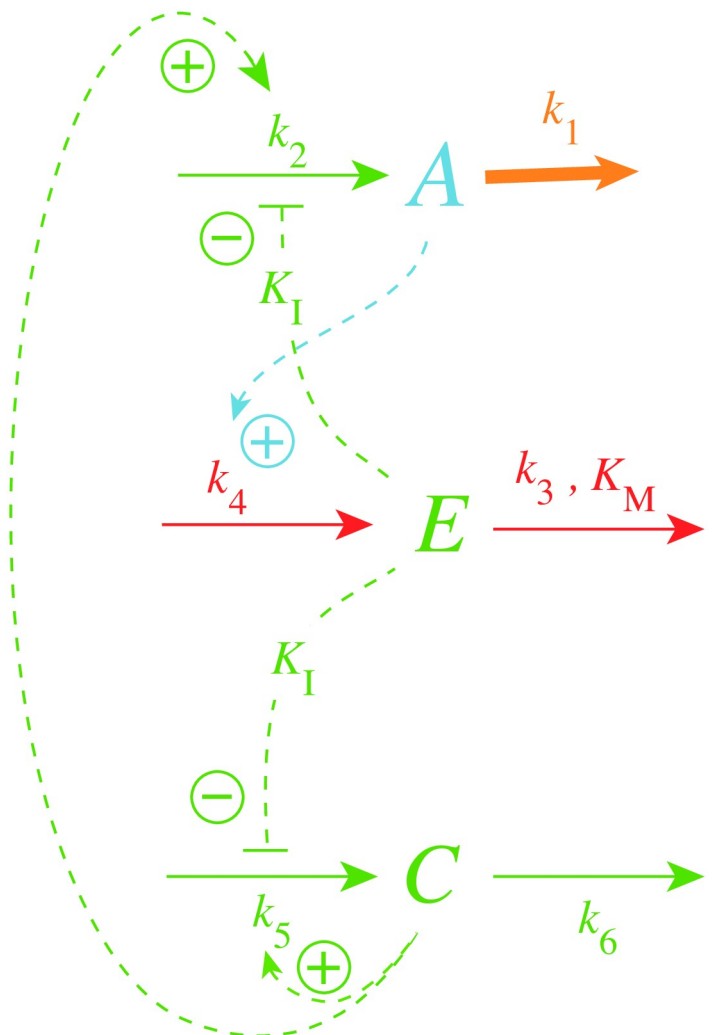

**Fig 7. Inclusion of autocatalysis in the generation of C.**

Note that for step-wise changes in $k_1$ $E$ becomes homeostatic controlled in addition to $A$, because $\dot{C} = 0$ in Eqs 9 or 10 implies that

$$E_{ss} = \left(\frac{k_5}{k_6} - 1\right) K_I \tag{11}$$

independent of the perturbation $k_1$.

**Comparing the influence of first-order and second-order autocatalysis in C on controller performance.** We have tested the influence of the first-order and second-order autocatalytic terms on the controller performance for $n = 4$. Considered were step-wise changes in $k_1$, together with linear, exponential, and hyperbolic increases of $k_1$. For all these perturbation types the controllers with both first-order and second-order autocatalysis show robust homeostasis and defend their set-points $A_{set} = k_3/k_4$ successfully. It is interesting to note that $E$ no longer decreases, but approaches a steady state during the time-dependent increase of $k_1$! We show here the results for exponentially and hyperbolically increasing $k_1$ values. The controller's behavior for step-wise and linear changes are described in S1 Text.

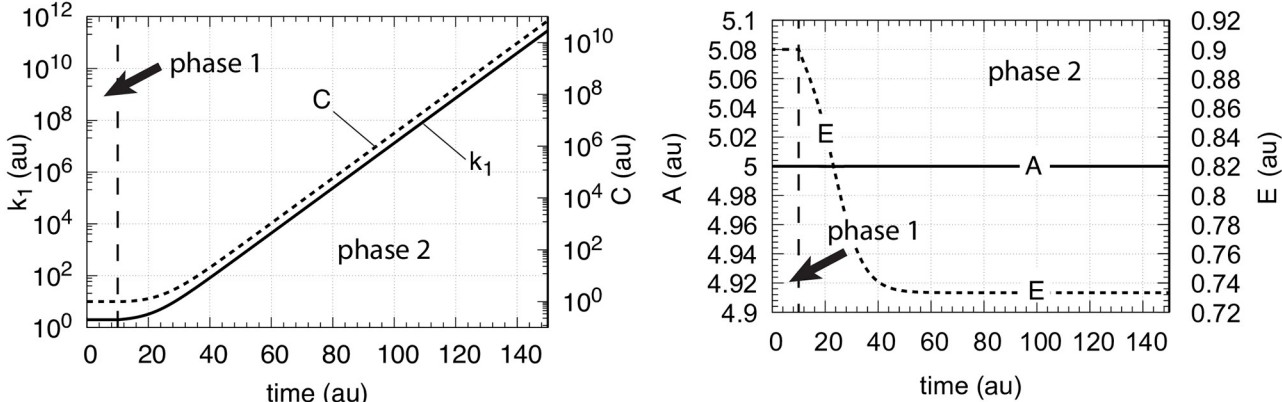

**Fig 8. Controller performance with first-order autocatalysis in C (Eq 9) and exponential increase of $k_1$.** Phase 1: the controller is at steady state at its set-point $A_{set} = 5.0$ with constant $k_1 = 2.0$. Initial concentrations: $A_0 = 5.0$, $E_0 = 0.9$, $C_0 = 1.0$. Phase 2: $k_1$ increases exponentially (left panel of Fig 3), $k_2 = 1 \times 10^5$, $k_3 = 5 \times 10^2$, $k_4 = 1 \times 10^2$, $k_5 = 10.0$, and $k_6 = 1.0$. $K_M = 1 \times 10^{-6}$, $K_I = 0.1$, $n = 4$ (Eq 6). Left panel: $k_1$ and $C$ as a function of time; right panel: $A$ and $E$ as a function of time.

**First-order autocatalysis in C.** The controller is described by Eqs 2, 6 and 9. The perturbation $k_1$ increases exponentially as described in the left panel of Fig 3.

Fig 8, left panel, shows that $C$ follows the exponential increase in $k_1$ closely, while the right panel shows that $E$ goes into a steady state with $A$ kept at its set-point $A_{set} = k_3/k_4 = 5.0$. The steady state value in $E$ can be calculated from Eq 9 by noting that this equation can be written as

$$\frac{1}{C} \cdot \frac{dC}{dt} = \frac{d \ln C}{dt} = \frac{k_5}{1 + \frac{E}{K_I}} - k_6 \tag{12}$$

Inserting into Eq 12 the value of d ln $C$/d$t$ (which is equal to $\dot{k}_1/k_1 = 0.2$ and using L'Hôpital's rule) together with the values of the other rate constants and solving for $E$, gives

$$E_{ss} = \left( \frac{k_5}{\frac{d \ln C}{dt} + k_6} - 1 \right) K_I = 0.7333 \tag{13}$$

in agreement with the numerical calculation.

One of the characteristic properties of the motif 2 derepression controller is its capability to tackle rapidly increasing perturbations, like hyperbolic growth [32]. While exponential growth in $k_1$ has a constant doubling time the doubling time in hyperbolic growth decreases exponentially and $k_1$ will (formally) reach infinity at a certain time point. We wondered whether the controller based on rate Eqs 2, 6 and 9 would still show this property.

In Eq 14 $k_1$ increases hyperbolically according to

$$k_1 = \frac{40.5}{\frac{40.5}{k_{1,p1}} - (t - t_{p1})} \tag{14}$$

where $k_{1,p1}$ is the constant value of $k_1$ during phase 1 ($k_1 = 2.0$ as in the previous calculations), while $t_{p1}$ is the duration of phase 1 (here 1 time unit). When time $t$ reaches 21.25 (the infinity limit) the value of $k_1$ goes formally to infinity. Fig 9 shows the behavior of the controller close to the infinity limit. In the calculation $t$ and $k_1$ reach 21.249997 and $1.4 \times 10^7$, respectively. During the last 0.25 time units $k_1$ increases by approximately 5-orders of magnitude. Despite

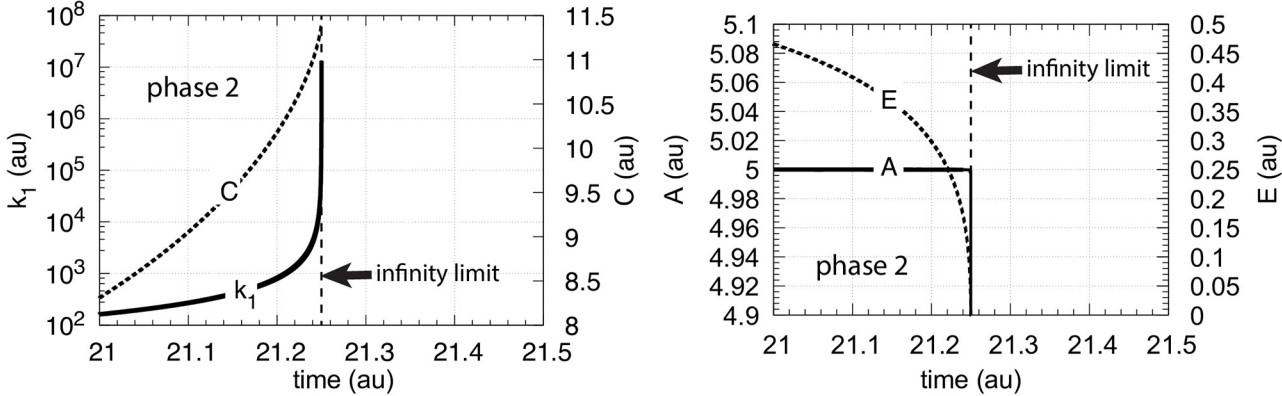

**Fig 9. Controller performance with first-order autocatalysis in *C* (Eq 9) and hyperbolic increase of $k_1$.** Phase 1 (not shown): the controller is at steady state at its set-point $A_{set}$ = 5.0 with constant $k_1$ = 2.0. Phase 1 lasts 1 time unit. Initial concentrations: $A_0$ = 5.0, $E_0$ = 0.9, $C_0$ = 1.0. Phase 2: $k_1$ increases in a hyperbolic fashion (Eq 14), $k_2 = 1 \times 10^5$, $k_3 = 5 \times 10^5$, $k_4 = 1 \times 10^5$, $k_5$ = 10.0, and $k_6$ = 1.0. $K_M = 1 \times 10^{-6}$, $K_I$ = 0.1, $n$ = 4 (Eq 6). Left panel: $k_1$ and *C* as a function of time just before $k_1$ reaches the infinity limit; right panel: corresponding *A* and *E* concentrations as a function of time.

this rapid increase in $k_1$ the controller is able to maintain homeostasis, but finally breaks down before the infinity limit is reached.

It is interesting to note that in case of hyperbolic growth the first-order autocatalytic growth in C (left panel, Fig 9) is not able to maintain a constant *E*, i.e., *E* decreases, but *A* is still kept at its set-point (right panel, Fig 9).

**Second-order autocatalysis in *C*.** We asked the question, how would the controller respond when the autocatalytic generation of *C* becomes second-order (Eq 10), with other words, when the generation of *C* is itself due to hyperbolic growth (S2 Text)?

Fig 10 shows the results when *C* is subject to second-order autocatalysis and $k_1$ increases exponentially. Although there is no apparent change in *A* in comparison with Fig 8 now *E* itself is under homeostatic control, besides *A*. *E*'s set-point can be calculated by setting Eq 10 to zero, which leads to

$$E_{set} = K_I \left( \frac{k_5}{k_6} - 1 \right) \tag{15}$$

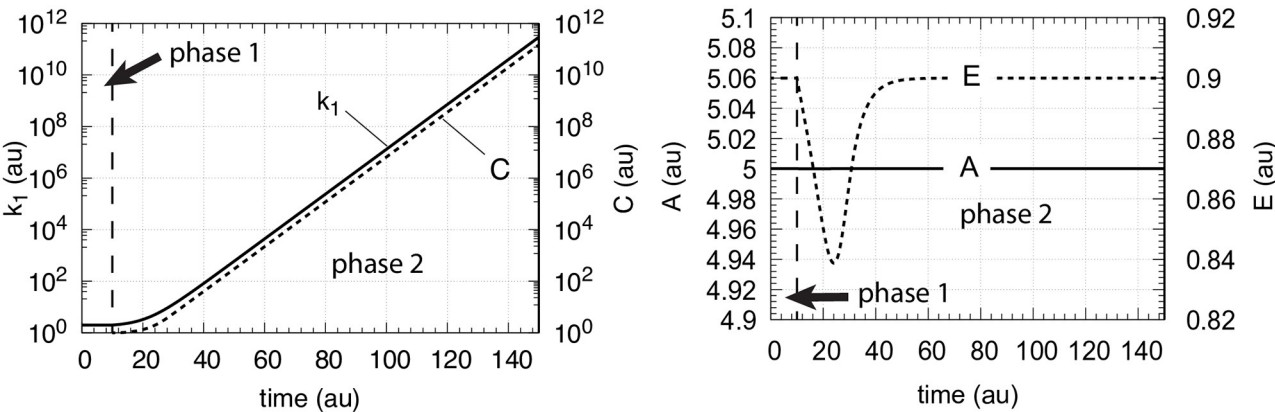

**Fig 10. Controller performance with second-order autocatalysis in *C* (Eq 10) and exponential increase of $k_1$.** Phase 1: the controller is at steady state at its set-point $A_{set}$ = 5.0 with constant $k_1$ = 2.0. Initial concentrations: $A_0$ = 5.0, $E_0$ = 0.9, $C_0$ = 1.0. Phase 2: $k_1$ increases exponentially (left panel of Fig 3), $k_2 = 1 \times 10^5$, $k_3 = 5 \times 10^2$, $k_4 = 1 \times 10^2$, $k_5$ = 10.0, and $k_6$ = 1.0. $K_M = 1 \times 10^{-6}$, $K_I$ = 0.1, $n$ = 4 (Eq 6). Left panel: $k_1$ and *C* as a function of time; right panel: corresponding *A* and *E* concentrations as a function of time.

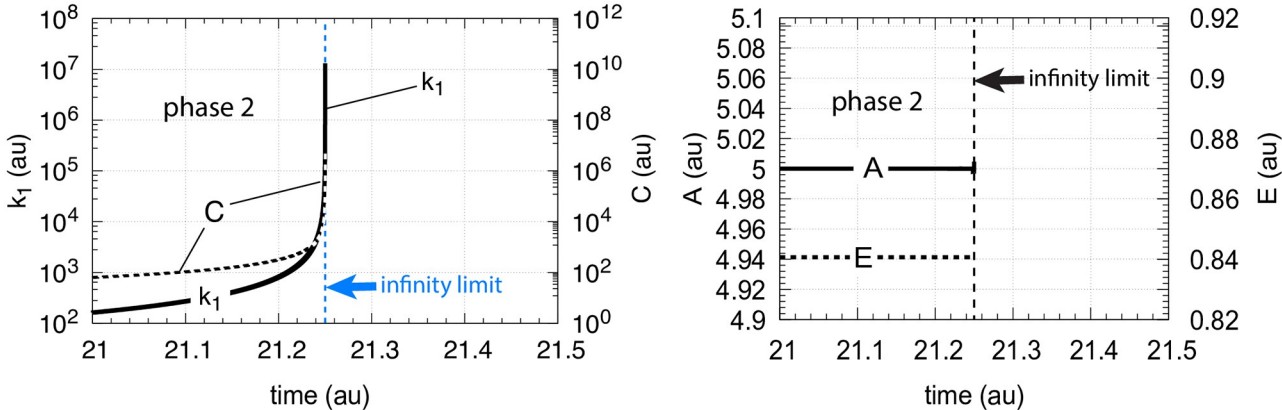

**Fig 11. Controller performance with second-order autocatalysis in C (Eq 10) and hyperbolic increase of $k_1$ (Eq 14).** Phase 1 (not shown): the controller is at steady state at its set-point $A_{set} = 5.0$ with constant $k_1 = 2.0$. Phase 1 lasts 1 time unit. Initial concentrations: $A_0 = 5.0$, $E_0 = 0.9$, $C_0 = 1.0$. Phase 2: $k_1$ increases hyperbolically. Rate constant values: $k_2 = 1 \times 10^5$, $k_3 = 5 \times 10^2$, $k_4 = 1 \times 10^2$, $k_5 = 10.0$, and $k_6 = 1.0$. $K_M = 1 \times 10^{-6}$, $K_I = 0.1$, $n = 4$ (Eq 6). Left panel: $k_1$ and $C$ as a function of time just before $k_1$ reaches the infinity limit (blue dashed line). At time 21.249997 $k_1 = 1.4 \times 10^7$, $C = 5.4 \times 10^6$. Right panel: $A$ and $E$ concentrations as a function of time.

Inserting the rate constant values from Fig 10 into Eq 15 gives a value for $E_{set}$ of 0.9 in agreement with the numerical values for $E_{ss}$.

Fig 11 shows the controller's behavior when $k_1$ increases hyperbolically with the same second-order autocatalysis as in Fig 10. However, while the controller is still able to keep $A$ at its theoretical set-point, the value of $E_{ss}$ shows now an offset below $E_{set}$ (Eq 15).

In the case when $k_1$ increases hyperbolically, we wondered how well motif 2 would perform without the help of $C$ in comparison with controllers that have first- or second-order autocatalysis in C?

An answer to this question is given in Fig 12, which shows the performance of a single motif 2 controller without $C$ in comparison with controllers having first- and second-order autocatalysis in $C$. Since the motif 2 controller without $C$ is subject to the limitation described by Eq 4, this controller breaks down earlier in comparison with those controllers having autocatalysis in $C$. The second-order autocatalytic controller performs best and keeps $A$ longest at its set-point before breakdown occurs near the infinity limit.

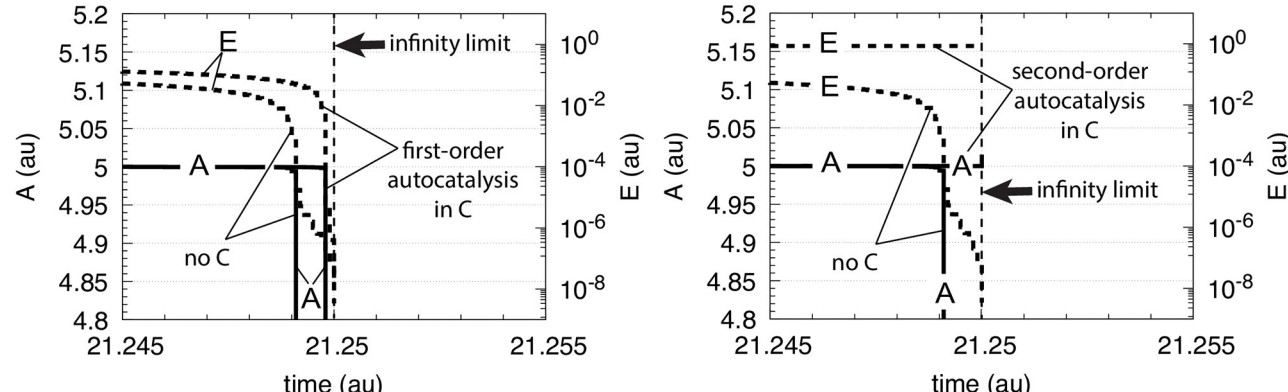

**Fig 12. Behavior of motif 2 controller (Fig 2) in comparison with a controller including first-order and second-order autocatalysis in C (Fig 7).** Rate constants for all three controllers: $k_2 = 1 \times 10^5$, $k_3 = 5 \times 10^5$, $k_4 = 1 \times 10^5$, $K_M = 1 \times 10^{-6}$, $K_I = 0.1$, $n = 4$. Additional rate constants for the autocatalytic controllers: $k_5 = 10.0$, and $k_6 = 1.0$. Initial concentrations: $A_0 = 5.0$, $E_0 = 0.9$, $C_0 = 1.0$ (when autocatalysis in $C$ is included).

## Oscillatory homeostats

This section is inspired by the fact that in physiology many cellular compounds or tissues show oscillatory behaviors [36–40], but are also under a homeostatic regulation. For example, circadian oscillations regulate hormones, blood glucose, and adapt organisms to the light/dark and seasonal changes on earth [41–46]. Another interesting example is the homeostatic stabilization of complex neural oscillations [47].

The motif 2 controller (Fig 2) becomes oscillatory when degradations with respect to $A$ and $E$ become zero-order. The resulting oscillator can maintain robust homeostasis [48] of $<A>$, where

$$< A >= \frac{1}{\tau} \int_0^\tau A(t)\, \mathrm{d}t \rightarrow A_{set} \tag{16}$$

and integration is taken over a certain time interval $\tau$.

This oscillator is identical to Goodwin's 1963 oscillator [49–51], although Goodwin was probably not aware of the oscillator's homeostatic property. The promotion of oscillations by zero-order kinetics have also been recognized by Kurosawa and Iwasa [52] in an alternative version of Goodwin's equations [53].

In case of motif 2, the oscillatory reaction scheme is shown in Fig 13 with the altered rate equation in $A$:

$$\dot{A} = \frac{k_2}{1 + \left(\frac{E}{K_I}\right)^n} - \frac{k_1 \cdot A}{K_M + A} \tag{17}$$

When $K_M \ll A$ and $n = 1$, Eqs 2 and 17 can be combined into a quasi-harmonic form [48]

$$\frac{\ddot{A}}{\frac{k_2 k_4 K_I}{(K_I + E)^2}} + A = A_{set} = \frac{k_3}{k_4} \tag{18}$$

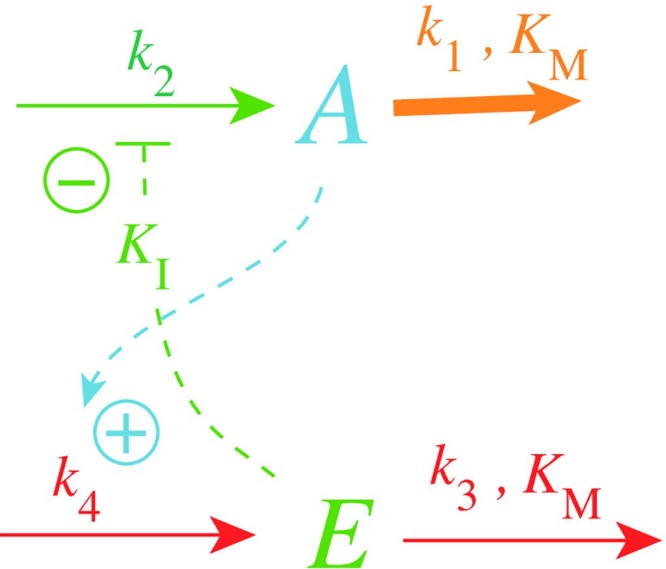

**Fig 13. Motif 2 becomes an oscillatory homeostat when** $K_M \ll E$ **(condition for integral control) and, in addition,** $K_M \ll A$. The resulting oscillator is identical to Goodwin's equations/oscillator from 1963 [49].

**Table 1. Harmonic and numerical periods.**

| $k_1$ | $k_2$ | $P_{harm}$ | $P_{num}$ |
|---|---|---|---|
| $9 \times 10^{-3}$ | $1 \times 10^{-2}$ | 22.0 | 22.1 |
| $9 \times 10^{-2}$ | 0.1 | 7.01 | 6.99 |
| 0.9 | 1.0 | 2.21 | 2.22 |

$K_I = 0.1$, $K_M = 1 \times 10^{-8}$, $k_2 = k_4 = 1.0$, $A_0 = 0.96$, $E_0 = 3.3 \times 10^{-3}$.

When the values of $k_1$ and $k_2$ are close to each other, the resulting oscillations are practically harmonic (sinusoidal) and the period $P$ of the oscillator can be estimated as

$$P_{harm} = \frac{2\pi}{\sqrt{\frac{k_2 k_4 K_I}{(K_I + <E>)^2}}} \tag{19}$$

with

$$<E> = \frac{1}{\tau} \int_0^\tau E(t)\, \mathrm{d}t \tag{20}$$

Table 1 shows three examples of numerically calculated periods $P_{num}$ in comparison with the corresponding harmonic periods $P_{harm}$. It may be noted that when $k_1$ and $k_2$ become significantly different, the resulting oscillations become highly nonlinear and $P_{harm}$ can only qualitatively indicate the period's dependence on rate constants and $<E>$.

Fig 14 shows, as a reference, the behavior of the oscillatory m2 controller (Fig 13) when $k_1$ increases exponentially. The value of $<A>$ is at the controller's set-point (5.0) and kept there, until, as in the non-oscillatory case (Fig 3), the controller breaks down when $E$ values become too low.

In the case when $C$ is included to improve controller performance (Fig 15), the resulting controller shows an increased lifetime. This is shown in Fig 16.

The presence of $C$ had no significant effect on the period (Fig 16, right panel) which decreased practically in the same manner as in Fig 14 when $C$ is absent. Despite the controller's increased lifetime in the presence of $C$, the controller will also in this case, due to the decrease in $E$, eventually break down as in the nonoscillatory case (Fig 6).

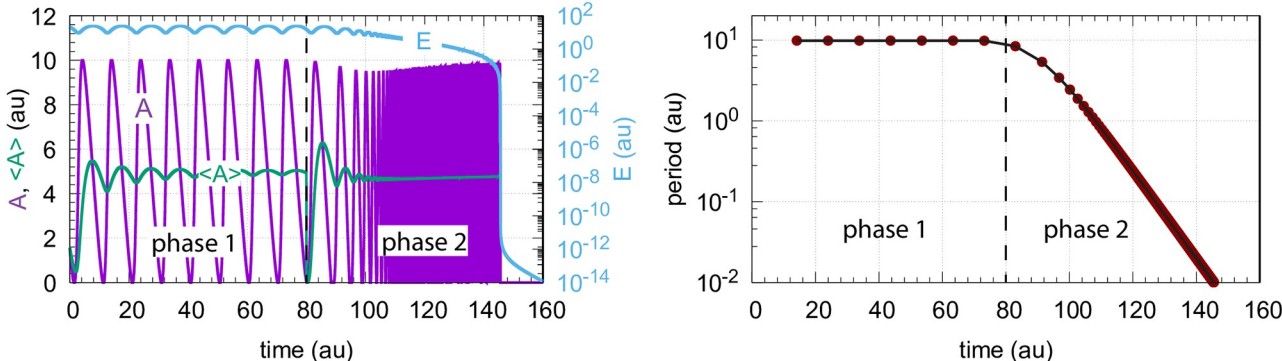

**Fig 14. Response of oscillatory motif 2 (Fig 13) when $k_1$ increases exponentially in phase 2 by the growth law described in the left panel of Fig 3.** Phase 1: $k_1$ is at 2.0; phase 2: At time t = 80 $k_1$ starts to increase exponentially. At the end of phase 2 $k_1$ is $1.8 \times 10^6$. Left panel: values of $A$, $<A>$, and $E$ as a function of time. Right panel: Calculated period as a function of time. Rate constants: $k_2 = 1.0 \times 10^5$, $k_3 = 5.0$, $k_4 = 1.0$, $K_I = 1.0$, $K_M = 1.0 \times 10^{-6}$, $n = 4$. Initial concentrations: $A_0 = 1.56$, $E_0 = 20.55$. The rate equations are given by Eqs 2 and 17.

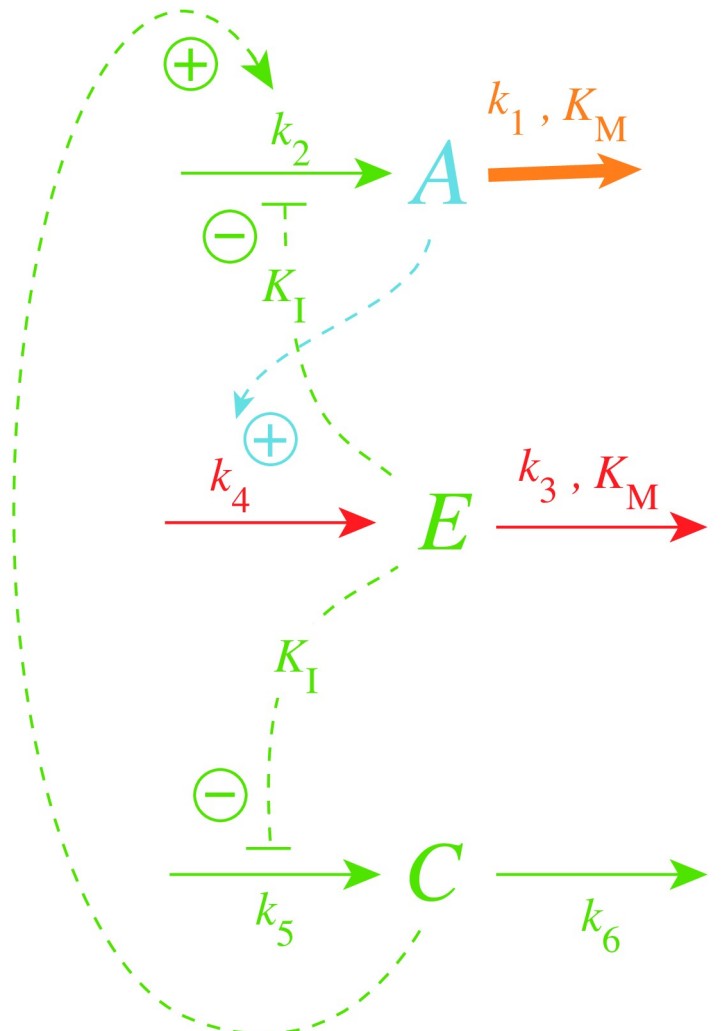

**Fig 15. Inclusion of $C$ in the oscillatory m2 controller.**

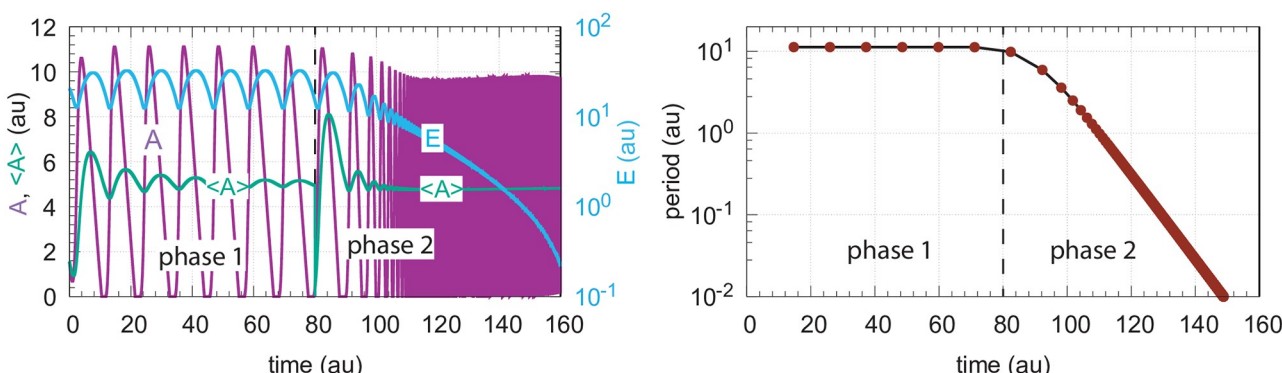

**Fig 16. Response of oscillatory motif 2 containing $C$ (Fig 15) when $k_1$ increases exponentially in phase 2 by the growth law described in the left panel of Fig 3.** Rate equations for $A$, $E$, and $C$ are given by Eqs 17, 2 and 7, respectively. Phase 1: $k_1$ is kept constant at 2.0; phase 2: At time t = 80 $k_1$ starts to increase exponentially. At the end of phase 2 $k_1$ is $1.8 \times 10^6$. Controller breakdown occurs just after 160 time units (data not shown). Left panel: values of $A$, $<A>$, and $E$ as a function of time. Right panel: Calculated period as a function of time. Rate constants: $k_2 = 1.0 \times 10^5$, $k_3 = 5.0$, $k_4 = 1.0$, $k_5 = 50.0$, $k_6 = 1.0$, $K_I = 1.0$, $K_M = 1.0 \times 10^{-6}$, $n = 4$. Initial concentrations: $A_0 = 1.56$, $E_0 = 20.55$, $C_0 = 1.0$.

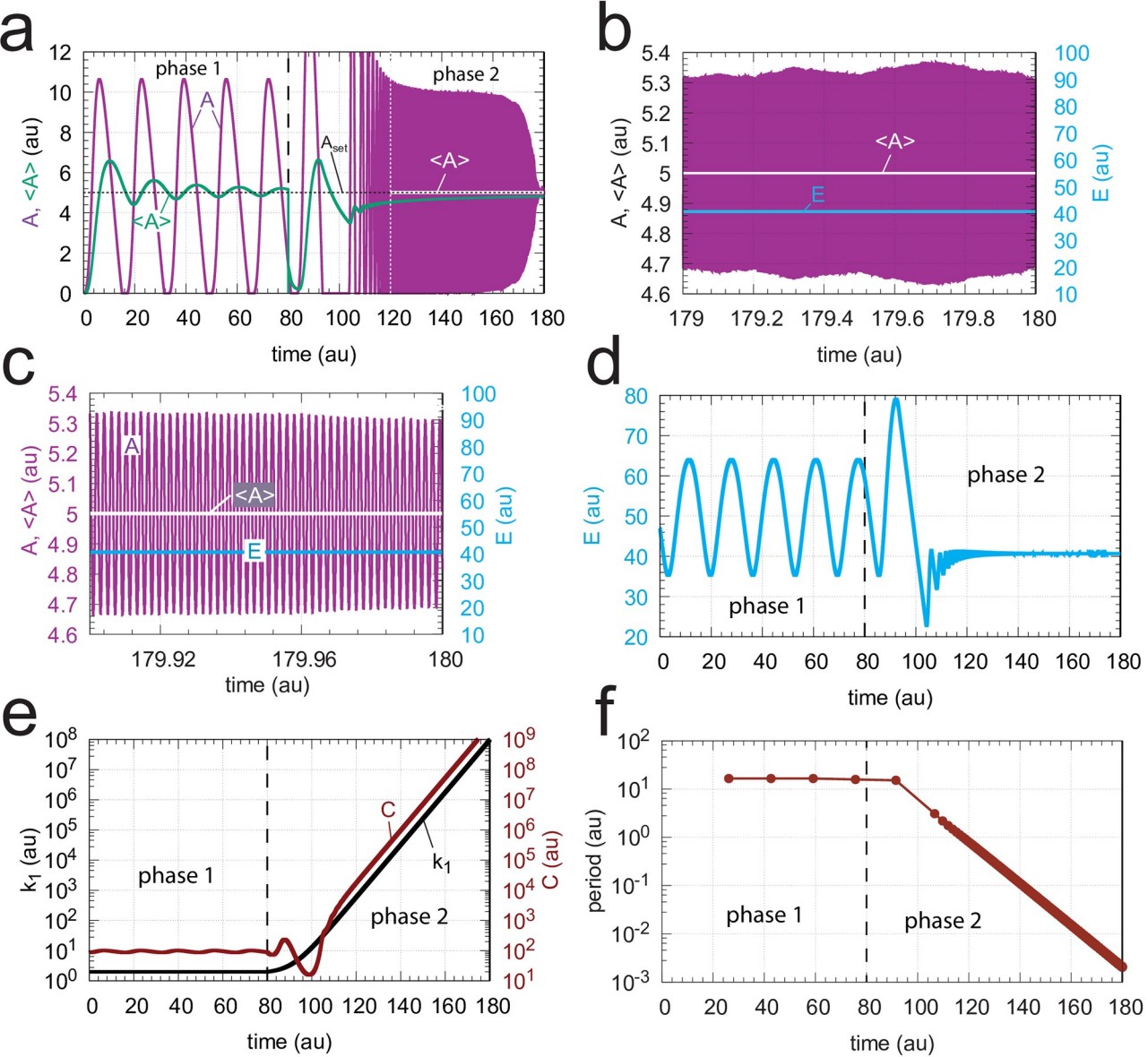

**Fig 17. Response of oscillatory m2 controller with first-order autocatalytic generation of $C$ and exponential increase of $k_1$.** Phase 1 (0-80 time units): $k_1 = 2.0$. Phase 2 (80-180 time units): $k_1$ increases exponentially as shown in the left panel of Fig 3. Rate equations are given by Eqs 17, 2 and 9. (a) $A$ (in purple) and overall $<A>$ (in green) as a function of time. The white outlined $<A>$ is the $<A>$ value calculated from 120 (white vertical line) to 180 time units showing that $<A> = A_{set} = 5.0$. (b) $A$, $<A>$, and $E$ for the time interval 179.0-180.0. (c) $A$, $<A>$, and $E$ for the time interval 179.9-180.0. (d) $E$ as a function of time. (e) $k_1$ and $C$ as a function of time. (f) The period as a function of time. Rate constants: $k_2 = 1 \times 10^5$, $k_3 = 5.0$, $k_4 = 1.0$, $k_5 = 5.0$ (phase 1), $k_5 = 50.0$ (phase 2), $k_6 = 0.1$ (phase 1), $k_6 = 1.0$ (phase 2), $K_I = 1.0$, $K_M = 1 \times 10^{-6}$. Initial concentrations: $A_0 = 2.684$, $E_0 = 61.55$, $C_0 = 86.21$, $n = 4$.

When $C$ is generated by first-order autocatalysis (described by Eq 9) the controller is able to defend $A_{set}$ for an extended time period (Fig 17a–17c) and keeps $<E>$ constant (Fig 17d). After an induction period, the controller is able to follow the exponentially increasing $k_1$ by $C$ (Fig 17e) and thereby becoming capable to defend $A_{set}$. Fig 17f shows the exponential decrease of the calculated period. Since in phase 2 $<E>$ is now kept at a constant level the controller will remain operative as long as $C$ can be increased and the activation of $A$ by $C$ is maintained.

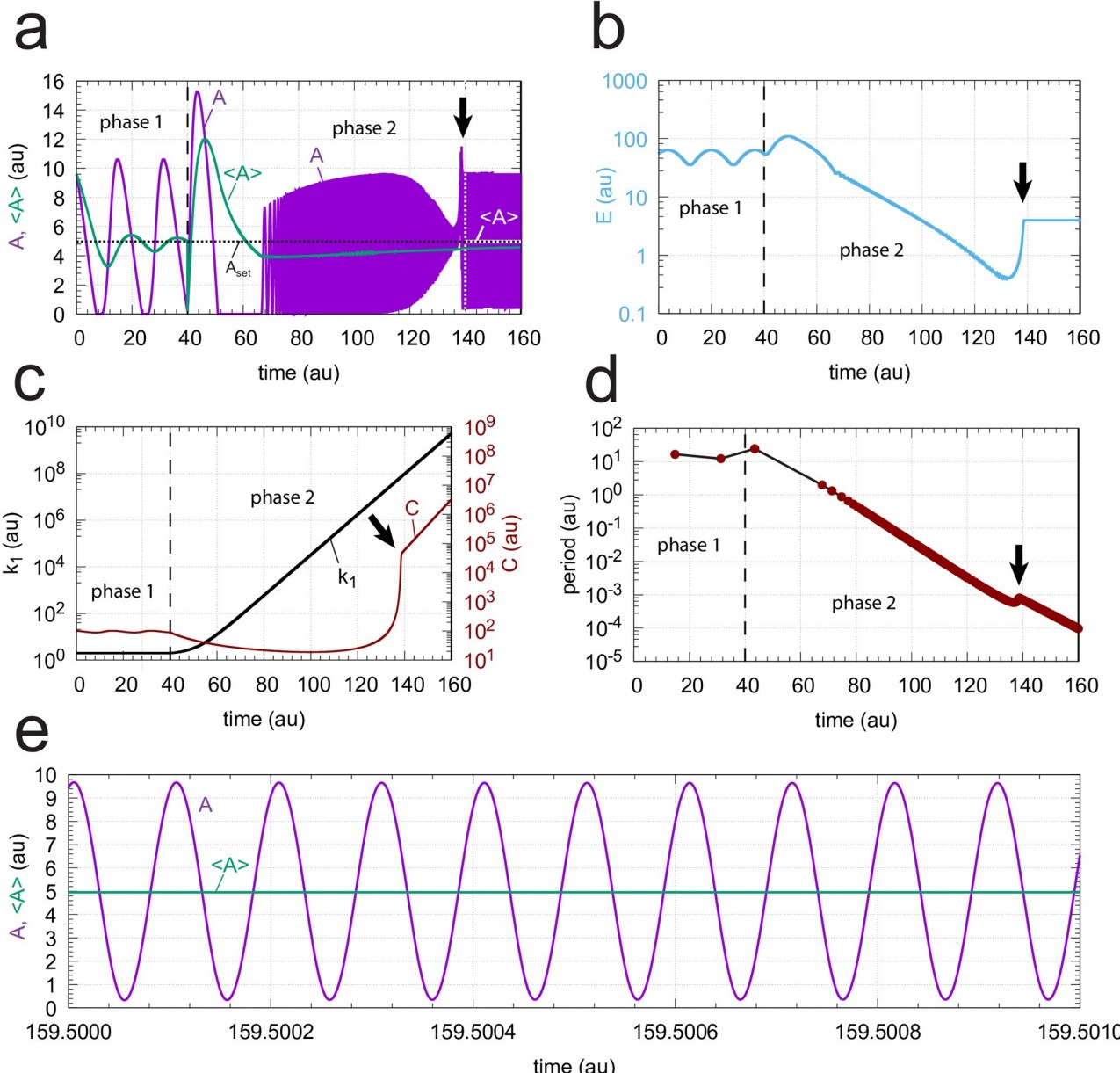

**Fig 18. Response of oscillatory m2 controller with second-order autocatalytic generation of $C$ and exponential increase of $k_1$.** Phase 1 (0-40 time units): $k_1 = 2.0$. Phase 2 (40-160 time units): $k_1$ increases exponentially as shown in the left panel of Fig 3. Rate equations are given by Eqs 17, 2 and 10. Downward arrows indicate the starting point when the controller is able to compensate for the increasing $k_1$ values. (a) $A$ (in purple) and overall $<A>$ (in green) as a function of time. The white outlined $<A>$ is the $<A>$ value calculated from 140 (white vertical line) to 160 time units showing that $<A> = A_{set} = 5.0$. (b) $E$ as a function of time. (c) $k_1$ and $C$ as a function of time. (d) The period as a function of time. (e) High frequency oscillations near the end of phase 2. $<A>$ is calculated for the time interval from 159.500 to 159.501 showing that $<A>$ (4.9989) is close to $A_{set} = 5.0$. Rate constants: $k_2 = 1 \times 10^5$ (phase 1), $k_2 = 1 \times 10^6$ (phase 2), $k_3 = 5.0$, $k_4 = 1.0$, $k_5 = 5 \times 10^{-2}$ (phase 1), $k_5 = 1 \times 10^{-3}$ (phase 2), $k_6 = 1 \times 10^{-3}$, $K_I = 1.0$, $K_M = 1 \times 10^{-6}$. Initial concentrations: $A_0 = 9.6$, $E_0 = 55.4$, $C_0 = 101.0$, $n = 4$.

When $C$ is generated by second-order autocatalysis (Eq 10) the resulting controller is, as for first-order autocatalysis, able to defend $A_{set}$. Fig 18 shows the case when $k_1$ increases exponentially. We found that an increase of $k_2$ by one order of magnitude during phase 2 was beneficial for the controller's homeostatic behavior. To avoid overcompensation, $k_5$ was decreased by one order of magnitude during phase 2. Fig 18a shows the time profiles of $A$ and $<A>$. Once

the controller is able to follow the increasing $k_1$ values (indicated by the downward arrows in the different panels) $<A>$ is at $A_{set}$, as indicated in panel a by the white outlined $<A>$.

In Fig 18b $E$ is shown as a function of time going into a steady state once control over the increasing $k_1$ values have been taken. The takeover of control is most clearly seen in Fig 18c when after the induction period $C$ is able to follow the increasing $k_1$. Fig 18d shows that the period is decreasing exponentially in line with the exponential increase of $k_1$. Fig 18e shows the high frequency oscillations in $A$ near the end of phase 2 having a period of approximately $10^{-4}$ time units. Calculating the $A$-average over the $10^{-3}$ time units shows that $<A>$ is at $A_{set} = 5.0$.

Finally we have tested a controller with second-order autocatalytic generation of $C$ when $k_1$ increases hyperbolically according to Eq 14. We found that a reduction of the unperturbed period in phase 1 to approximately 2 time units gave a good illustration of an operational controller under these conditions. The reduced period was achieved by increasing $k_4$ to 50.0. This led to a decrease in $A_{set}$ (= $k_3/k_4$) to 0.1. Fig 19a shows a semilogarithmic plot of $A$ and $<A>$ as a function of time. At time t = 50.0 $k_1$ starts to grow (Eq 14) with $k_{p,1} = 2.0$. The infinity limit is reached at 70.25 time units. Also here we observe an induction period in which the controller adapts to the increasing $k_1$. At about 60 time units the controller is able to oppose the increasing $k_1$ values. At the same time $<E>$ goes into a steady state (Fig 19b) and $C$ is able to follow $k_1$ (Fig 19c). The period decreases in a corresponding manner as $k_1$ increases (Fig 19d), and the controller is able to defend $A_{set}$. Fig 19e shows that $<A> = 0.0997$ when $<A>$ is determined between 60 time units and close to the infinity limit.

## Period homeostasis with respect to step-wise perturbations in $k_1$

Since in some oscillatory physiologies, like circadian rhythms, period homeostasis is observed with respect to certain step-wise environmental perturbations, for example in temperature or pH [54, 55], we wondered whether it would be possible to include an additional variable to one of the above oscillatory controllers which would give homeostasis not only in $<A>$ but also in the oscillator's period. We have previously shown [48] how the basic oscillatory m2 motif (Fig 13) can show period homeostasis by the addition of controller variables that keep $E$ and the chemical fluxes through $E$ at a constant level. Here we show that we can do the same by taking, as an example, the controller described in Fig 15 (including autocatalytic formation of $C$).

Fig 20 shows the controller's reaction scheme with the additional variable $I_1$, which keeps $<C>$ and thereby $<E>$ under homeostatic control. The rate equations are:

$$\dot{A} = \frac{k_2 C}{1 + \left(\frac{E}{K_I}\right)^n} - \frac{k_1 \cdot A}{K_M + A} + k_9 \cdot I_1 \tag{21}$$

$$\dot{E} = k_4 \cdot A - \frac{k_3 \cdot E}{K_M + E} \tag{22}$$

$$\dot{C} = \frac{k_5 C}{1 + \frac{E}{K_I}} - k_6 \cdot C \tag{23}$$

$$\dot{I_1} = k_7 \cdot C - \frac{k_8 \cdot I_1}{K_M + I_1} \tag{24}$$

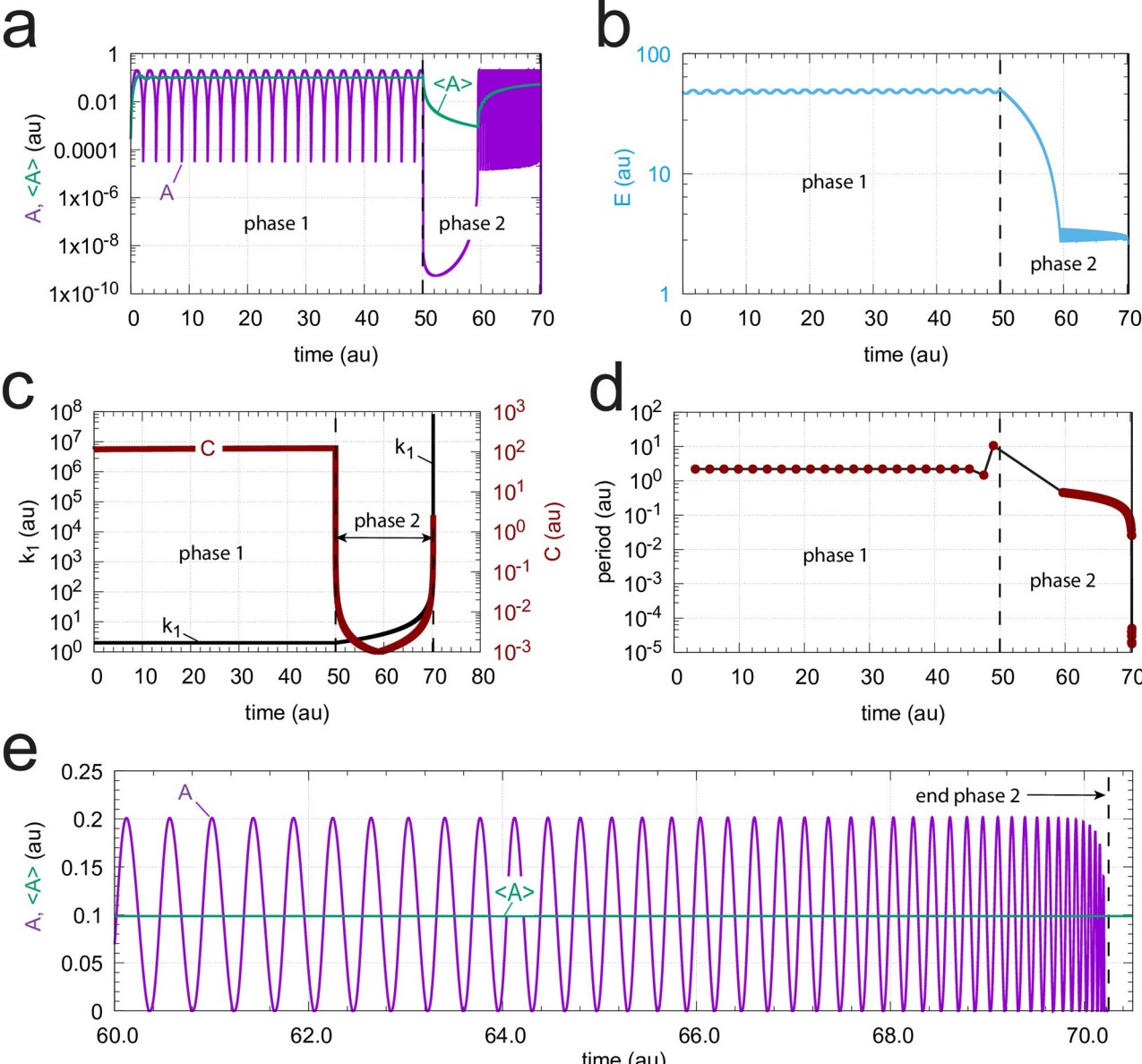

**Fig 19. Response of oscillatory m2 controller with second-order autocatalytic generation of *C* and hyperbolic increase of $k_1$.** Rate equations are given, as in Fig 18, by Eqs 17, 2 and 10. Phase 1 (0-50 time units): $k_1 = 2.0$. Phase 2 (50-70.24999992 time units) $k_1$ increases hyperbolically (Eq 14) from 2.0 to $5.8673 \times 10^8$. $A_{set} = 0.1$. (a) *A* and average $<A>$ as a function of time. $A_{set} = 0.1$. (b) Concentration of *E* as a function of time. (c) $k_1$ and *C* time profiles. (d) The period as a function of time. (e) Concentration of *A* and calculated $<A>$ (= 0.0997) during phase 2 (time interval 60.0-70.24999992) when oscillations are present. Rate constants: $k_2 = 1 \times 10^5$ (phase 1), $k_2 = 1 \times 10^6$ (phase 2), $k_3 = 5.0$, $k_4 = 50.0$, $k_5 = 5 \times 10^{-2}$ (phase 1), $k_5 = 1 \times 10^3$ (phase 2), $k_6 = 1 \times 10^{-3}$ (phase 1), $k_6 = 1.57 \times 10^2$ (phase 2), $K_I = 1.0$, $K_M = 1 \times 10^{-6}$. Initial concentrations: $A_0 = 2.736 \times 10^{-4}$, $E_0 = 4.793 \times 10^1$, $C_0 = 1.489 \times 10^2$, $n = 4$.

$I_1$ acts as an additional inflow controller with the property to keep $<C>$ at a constant level. The two compensatory fluxes

$$j_2 = \frac{k_2 C}{1 + \left(\frac{E}{K_I}\right)^n} \qquad (25)$$

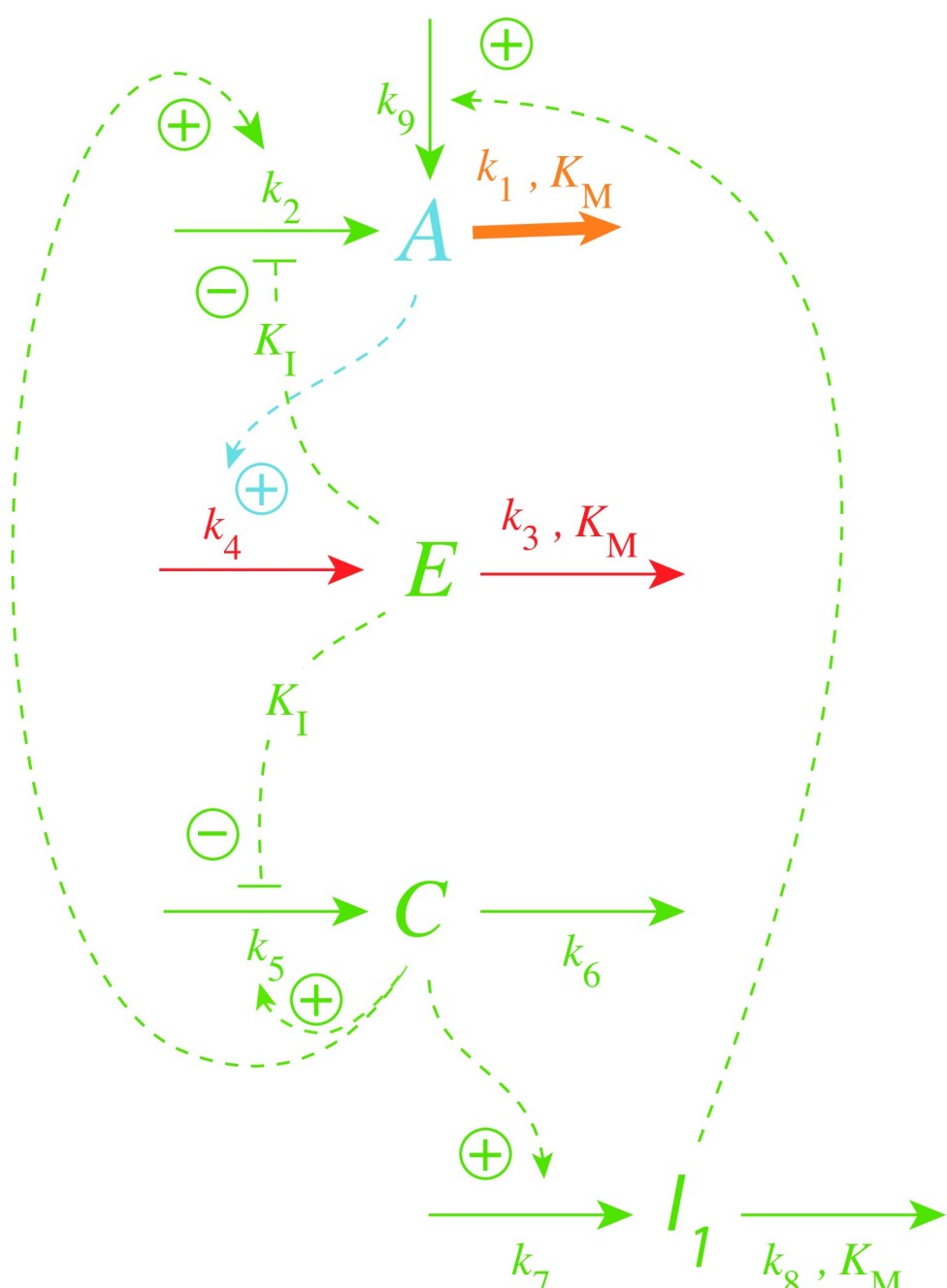

**Fig 20. Reaction scheme of oscillatory m2 controller with autocatalytic generation of $C$ and period homeostasis with respect to step-wise perturbations in $k_1$.**

and

$$j_9 = k_9 \cdot I_1 \tag{26}$$

act together such that $<A>$, $<E>$, and $<C>$ are under homeostatic control, which leads to regulated fluxes through $A$, $E$, and $C$ such that the period of the oscillator becomes constant.

Fig 21 illustrates the period homeostasis for step-wise changes of $k_1$ from 2.0 to 10.0. The period length during phase 1 ($k_1 = 2.0$) is approximately 18 time units. Directly after the step

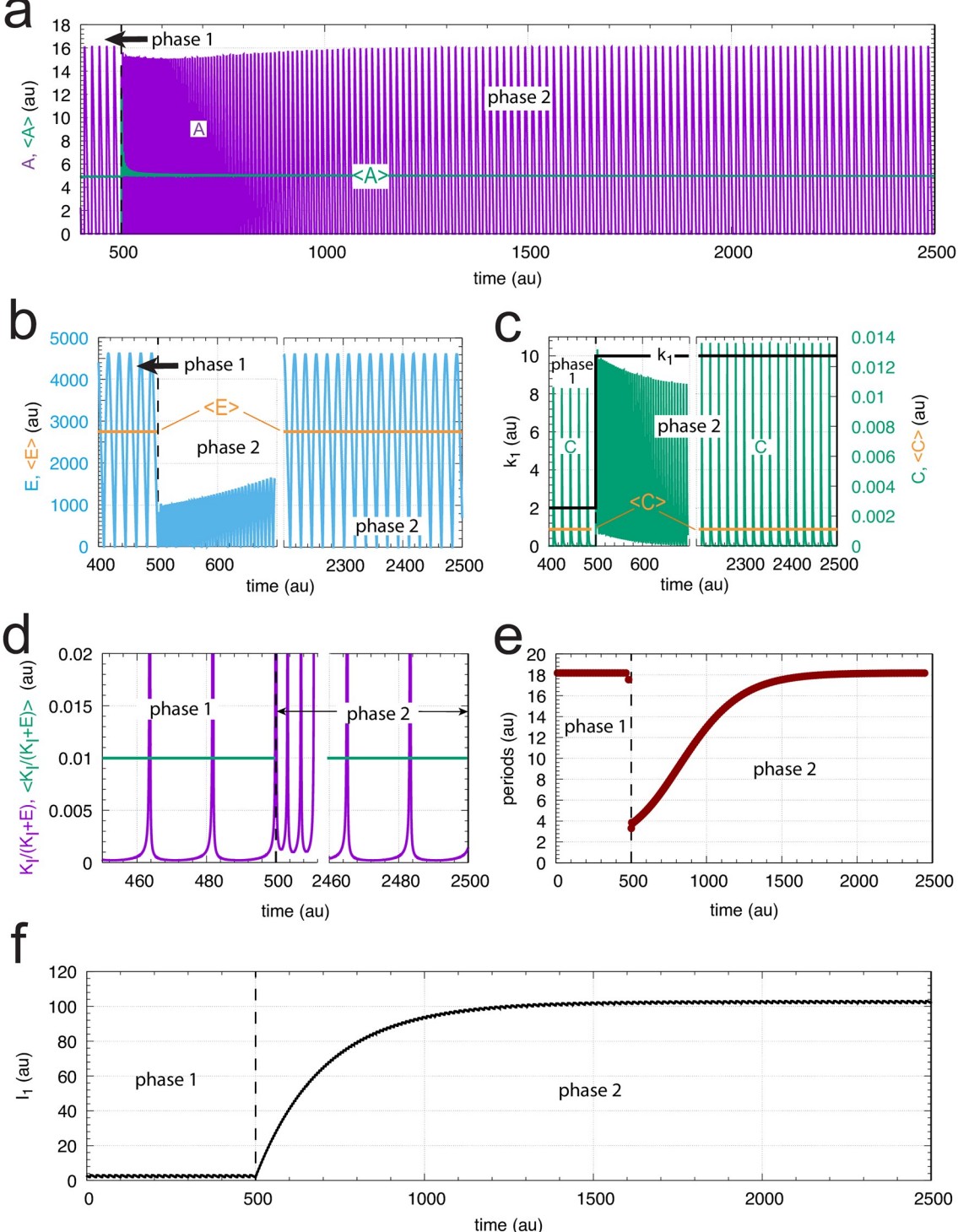

**Fig 21. Example of period homeostasis of the controller in Fig 20 when $k_1$ changes from 2.0 (phase 1) to 10 at time $t$ = 500.** Panel a shows oscillatory $A$ and the calculated average $<A>$ as a function of time. (b) Concentration of $E$ as a function of time (in blue) and $<E>$ (orange lines). For phase 1 $<E>$ was calculated for the entire phase, i.e., for the time interval 0-500, while in phase 2 $<E>$ was calculated for the time interval 2000-2500. (c) Step-wise change of $k_1$ from 2 to 10 (left ordinate, outlined in black). Right ordinate: concentration of C as a function of time. Orange lines: calculated average $<C>$ for the time intervals given in panel b. (d) Time plot (in purple) and average value (in green) of respectively $K_I/(K_I + E)$ and $<K_I/(K_I+E)>$. The average values are calculated for the time intervals as stated for panel b. For visibility, ordinate values, which have maximum values of 1 are cut off at 0.02. (e) Calculated period length as a function of time showing that $I_1$ manages to keep period homeostatically controlled. (f) $I_1$ as a function of time. Rate

constants: $k_2 = 1 \times 10^5$, $k_3 = 500$, $k_4 = k_5 = 100$, $k_6 = 1.0$, $k_7 = 100$, $k_8 = 0.1$, $k_9 = 8 \times 10^{-2}$, $K_I = 1.0$, $K_M = 1 \times 10^{-6}$. Initial concentrations: $A_0 = 2.683$, $E_0 = 4.463 \times 10^3$, $C_0 = 4.108 \times 10^{-6}$, $I_{1,0} = 2.705$, $n = 1$.

in $k_1$ the period drops to approximately 4 time units but then increases to its original value due to the increase in $I_1$ (panels a, e, and f). The set-point of $<C>$ can be calculated from Eq 24 by setting $< \dot{I}_1 > = 0$, which leads to

$$< \dot{I}_1 > = < k_7 \cdot C > - \left\langle \frac{k_8 \cdot I_1}{K_M + I_1} \right\rangle = 0 \quad \Rightarrow \quad < C > = \frac{k_8}{k_7} \tag{27}$$

by assuming that $<I_1/(K_M + I_1)> = 1$, i.e, $K_M \ll I_1$.

Fig 21c shows that the calculated $<C>$ value ($= 0.001$) is in perfect agreement with Eq 27. Applying $< \dot{C}/C > = 0$ in Eq 23 leads to the condition

$$\left\langle \frac{K_I}{K_I + E} \right\rangle = \frac{k_6}{k_5} = 0.01 \tag{28}$$

which is obeyed, as seen in Fig 21d. However, despite the fact that Eq 28 is fulfilled, we were not able to extract an analytical value for $<E>$. To get $<E>$ (Fig 21b) we used the numerical solution of the rate equations.

An interesting aspect is whether period homeostasis can also be obtained when the perturbation becomes time-dependent. We will deal with this situation more generally in another paper.

## Controller breakdown due to saturation

We have shown that the breakdown of the simple derepression m2 controllers (Figs 2 and 13) can be delayed or even circumvented by including a component $C$ which is activated by derepression from $E$, but itself activates the generation of $E$ via $A$ (Fig 5). However, there is the question how saturation may affect the performance of the controllers. For example, although the controllers containing a first-order autocatalysis in $C$ are apparently able to follow exponentially increasing $k_1$ values (Figs 9 and 17) they eventually will break down since neither the perturbation $k_1$ nor the increase in $C$ can continue ad infinitum.

In this respect, the models presented here need to be considered as idealizations. For example, concerning the growth of $k_1$ and $C$, $k_1$ and $C$ will eventually approach saturation levels. In the case of $k_1$, the removal of $A$ may be due to an enzyme or a transporter, which eventually will go into saturation. Likewise, $C$ will be generated by corresponding enzymatic processes and will be subject to saturation (see for example Ref [18] (Supporting Material) and Ref [56]).

A brief overview over these breakdown scenarios are now given. When $k_1$, i.e. the removal of $A$, goes into saturation before the production of $C$ and before the $C$-signaling to the compensatory flux, the controller will be able to keep homeostasis of $A$ at its set-point, at a high but constant level of $k_1$. When $C$-signaling goes into saturation before $k_1$ and $C$ become saturated, then the controller will break down due to an unbalanced exponential increase of $C$. Finally, when $C$ production becomes saturated, but the removal of $A$ by $k_1$ and $C$-signaling are still operative, then breakdown of the controller occurs, because the kinetics of the $C$ production are not able to oppose the rapid decrease of $A$ and $A$ levels will go to zero.

## Example: The TOC1/PRR5-RVE8 negative feedback loop

A question natural to ask, is whether the above "A-E-C" regulatory circuits can be found in physiology. Would the properties be the same as in the isolated case, i.e., as studied here? Since

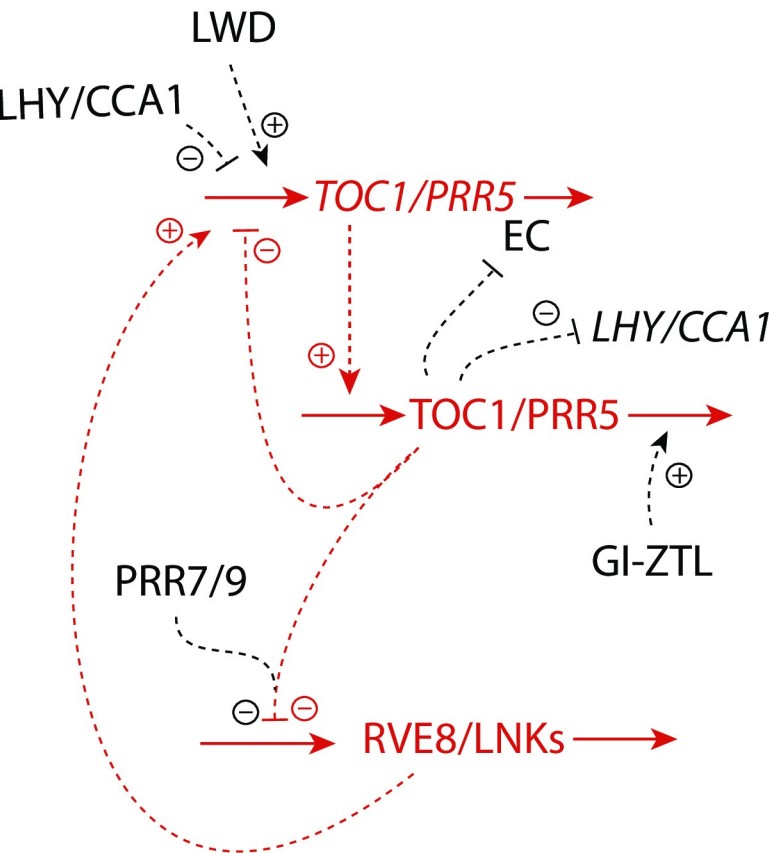

**Fig 22. Part of the plant circadian clock involving the *TOC*1 m2-type feedback loop.** The A-E-C motif (Fig 5) is outlined in red.

circadian rhythms are based on transcriptional-translational negative feedback loops, the Goodwin oscillator and the m2-scheme has served as a basic model to describe circadian oscillations [49–51, 57, 58]. We have taken the plant circadian clock organization and looked for the A-E-C motif (Fig 5) there. In plants the circadian organization is complex [59, 60] and consists of several interlocked negative feedback loops where each of them can approximately be described by a basic m2-scheme (Fig 13). Plants have a morning oscillator based on the genes *LHY* (Late Elongated Hypocotyl) and *CCA*1 (Circadian Clock Associated 1) and an evening complex (EC) which also contains an autoregulatory negative feedback loop. In addition, there appear to be transcriptional-translational negative feedbacks in the organization of pseudo-response regulators (*PRR*'s). The *PRR* gene family consists of five paralogue genes (*PRR*1, 3, 5, 7, and *PRR*9). *PRR*1 (also known as *Timing of CAB expression 1 (TOC1)*) is presently one of the best characterized gene in the *PRR* family. They are implicated in clock function and act as period controlling factors [61]. *TOC*1 and *PRR*5 are interlocked with the morning oscillator components *LHY* and *CCA*1 and the evening complex. It is in the *TOC*1/*PRR*5 feedback structure including the *RVE*8 (*REVEILLE*8) gene we find the A-E-C motif.

Fig 22 shows part of the plant circadian network including *TOC*1/*PRR*5 and *RVE*8. The *TOC*1 and *RVE*8 genes, when mutated, affect the period of the plant circadian clock. In addition, *RVE*8 has also an influence on the circadian period with respect to temperature (temperature compensation) [62]. *RVE*8 interacts with *LHY*, *CCA*1, and the EC, which in their turn also have an influence on the plant circadian rhythm. The *TOC*1/*PRR*5-mRNA's are part of a

negative feedback loop where the proteins TOC1 and PRR5 feed negatively back on their transcription. RVE8 is a MYB-like transcription factor and activates the transcription of *TOC*1/*PRR*5, but needs NIGHT LIGHT-INDUCIBLE AND CLOCK-REGULATED 1 and 2 (LNK1 and LNK2) [63, 64] to do that. TOC1/PRR5 on their side inhibit the production of RVE8 possibly by transcriptional repression [65]. Our results with the A-E-C motif suggests, in agreement with experimental implications [62], that *RVE*8 may take part in the homeostasis of the *TOC*1/*PRR*5 negative feedback loop, to stabilize homeostatic properties of the plant circadian clock, including the period. For example, overexpression of RVE8 leads to a shorter circadian period. In analogy, increase of $k_5$ in Fig 5 leads also to a shorter period. However, the TOC1/PRR5 circuits are highly interlocked with other clock components. Thus, other roles of the *TOC*1/*PRR*5-*RVE*8 loop may emerge when detailed models of the plant circadian clock are considered.

While we started to find an improvement of the m2-regulatory loop, we arrived at the A-E-C motif. We feel that this or similar feedback structures may be found in other physiological regulations, but more investigations are needed in this respect.

## Supporting information

**S1 Matlab. Matlab programs.** A zip-file with Matlab programs showing results from Fig 4 ($n$ = 4), Fig 6 ($n$ = 4, i = 1), Figs 8, 10 and 14.
(ZIP)

**S1 Text. Controller performance towards step-wise changes in $k_1$ and linearly increasing $k_1$ values.** Comparison between controllers (Fig 7) having first-order and second-order autocatalytic generation of *C*.
(PDF)

**S2 Text. Hyperbolic growth by higher-order autocatalysis.** It is shown that autocatalysis with an order larger than one shows hyperbolic growth.
(PDF)

## Author Contributions

**Conceptualization:** Peter Ruoff.

**Data curation:** Qaiser Waheed, Peter Ruoff.

**Formal analysis:** Gorana Drobac, Qaiser Waheed, Peter Ruoff.

**Investigation:** Gorana Drobac, Qaiser Waheed, Behzad Heidari, Peter Ruoff.

**Methodology:** Qaiser Waheed, Peter Ruoff.

**Project administration:** Qaiser Waheed, Peter Ruoff.

**Software:** Qaiser Waheed, Peter Ruoff.

**Supervision:** Peter Ruoff.

**Validation:** Gorana Drobac, Qaiser Waheed, Behzad Heidari, Peter Ruoff.

**Visualization:** Gorana Drobac, Qaiser Waheed, Behzad Heidari, Peter Ruoff.

**Writing – original draft:** Peter Ruoff.

**Writing – review & editing:** Qaiser Waheed, Behzad Heidari.

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
