## [Decision Letter · Decision Letter 0]

11 Dec 2020

PONE-D-20-32544

An amplified derepression controller with multisite inhibition and positive feedback

PLOS ONE

Dear Dr. Ruoff,

Thank you for submitting your manuscript to PLOS ONE. After careful consideration, we feel that it has merit but does not fully meet PLOS ONE’s publication criteria as it currently stands. Therefore, we invite you to submit a revised version of the manuscript that addresses the points raised during the review process.

We look forward to receiving your revised manuscript.

Kind regards,

Jae Kyoung Kim

Academic Editor

PLOS ONE

Journal Requirements:

2.) We note that you have included the phrase “data not shown” in your manuscript. Unfortunately, this does not meet our data sharing requirements. PLOS does not permit references to inaccessible data. We require that authors provide all relevant data within the paper, Supporting Information files, or in an acceptable, public repository. Please add a citation to support this phrase or upload the data that corresponds with these findings to a stable repository (such as Figshare or Dryad) and provide and URLs, DOIs, or accession numbers that may be used to access these data. Or, if the data are not a core part of the research being presented in your study, we ask that you remove the phrase that refers to these data.

Reviewers' comments:

Reviewer's Responses to Questions

**Comments to the Author**

1. Is the manuscript technically sound, and do the data support the conclusions?

Reviewer #1: Yes

Reviewer #2: Yes

2. Has the statistical analysis been performed appropriately and rigorously? 

Reviewer #1: N/A

Reviewer #2: N/A

3. Have the authors made all data underlying the findings in their manuscript fully available?

Reviewer #1: Yes

Reviewer #2: Yes

4. Is the manuscript presented in an intelligible fashion and written in standard English?

Reviewer #1: Yes

Reviewer #2: Yes

5. Review Comments to the Author

Reviewer #1: This paper describes approach to address issue that arise when derepression encounters low inhibitor concentration. The approach is interesting and would generate interest in particular among control and synthetic biology community. However there are several major points that need to be address. See my comments below.

1) Page 8, Line 35: It seems rather abrupt and sudden to mention motif 2 from Ref [18] without providing proper motivation on why this motif is of particular interest compared to other motifs in Ref [18]. Some discussion would be great.

2) Equation (1) and (2). Here is a suggestion: The authors should consider changing the use of variable E to U to (i) avoid potential associating E with Error and (ii) U is commonly known in control community to represent control action/signal. As a note, it took me quite a while to realise that E is not error but control action, which left me initially quite confused.

3) Page 8, Line 47. The assumption made here is that K_M << E. It seems to me that the overall numerical simulation is carried out with K_M << E and it gives the impression the mechanism of Motif 2 will work based on this assumption. What if KM is not significantly smaller than E? How would this affect the overall analysis and conclusion? Has this point being considered as there is no certainty that in practice, K_M << E often hold.

4) Page 9, Line 53. The word “breaks” should be brakes.

5) The overall figure quality needs to be improved. There is element of blurring/smearing making the reading of the figure difficult.

6) Figure 3: What determines the choice of parameters for the Equation of k1 shown in left panel of Figure 3? In practice, how common is k1 be subjected to this exponential type of perturbation and is it common that k1 having such large value, in the range up to 10^7.

7) Page 10: Line 89-90: Since you have mentioned this - how would the performance of using different value of KI be? Has this being investigated? And how different should this value of KI should be?

8) Page 11, Equation (6), (2) and (7).: This is another suggestion: It is quite odd to label equation this manner where (2) is appearing between (6) and (7). I understand it is the same equation but it is rather odd. Why not just label them as (6)-(8) in a sequential manner?

9) Page 15, Line 225 and also across the article starting from here: What is this m2 controller? It hasn’t been defined earlier.

10) Page 20, Line 340: Can the author clarifies this statement that RVE8 interacts with LHY/CCA1? I don't think that statement is correct as RVE8 does not interact with LHY/CCA1 as shown in the two articles below. If the authors look at the interaction of plant circadian genes shown in Figure 1 of both respective articles, there is no depiction of RVE8 interacting with LHY/CCA1.

[1] Fogelmark, Karl, and Carl Troein. "Rethinking transcriptional activation in the Arabidopsis circadian clock." PLoS Comput Biol 10.7 (2014): e1003705.

[2] Foo, Mathias, David E. Somers, and Pan-Jun Kim. "Kernel architecture of the genetic circuitry of the Arabidopsis circadian system." PLoS computational biology 12.2 (2016): e1004748.

Reviewer #2: This manuscript is well-written and very interesting to read.

The introduction of the paper provides a nice description of the flow chart of control theory from the era of mechanical control to applications for physiology and biology.

It should also be appreciated that the authors introduced different control features (such as multisite inhibition, a new variable $C$ or $I$, and the positive feedback of $C$) in a stage-by-stage manner in order to show how the control motif can be improved. They also provide schematic figures and plots that can substantially help readers to easily follow the outline of the paper. Despite the well-elaborated main flow of the paper, more mathematical proof or intuitions need to be provided to explain why such additional control features will be required to achieve the aim of the controller. Therefore I would like to suggest the following major and minor revisions before publication.

Please see the attached file for the review comments.

6. PLOS authors have the option to publish the peer review history of their article (what does this mean?). If published, this will include your full peer review and any attached files.

Reviewer #1: No

Reviewer #2: No

<gdiv></gdiv>

---

## [Author Response · Author response to Decision Letter 0]

30 Dec 2020

Please, see attached file "Response to Reviewers"

---

## [Decision Letter · Decision Letter 1]

19 Jan 2021

An amplified derepression controller with multisite inhibition and positive feedback

PONE-D-20-32544R1

Dear Dr. Ruoff,

We’re pleased to inform you that your manuscript has been judged scientifically suitable for publication and will be formally accepted for publication once it meets all outstanding technical requirements. Importantly,  please go over additional minor comments by reviewers (see below and attached files) and revise the manuscript accordingly during the proof process. 

Kind regards,

Jae Kyoung Kim

Academic Editor

PLOS ONE

Additional Editor Comments (optional):

Reviewers' comments:

Reviewer's Responses to Questions

**Comments to the Author**

1. If the authors have adequately addressed your comments raised in a previous round of review and you feel that this manuscript is now acceptable for publication, you may indicate that here to bypass the “Comments to the Author” section, enter your conflict of interest statement in the “Confidential to Editor” section, and submit your "Accept" recommendation.

Reviewer #1: All comments have been addressed

Reviewer #2: All comments have been addressed

2. Is the manuscript technically sound, and do the data support the conclusions?

Reviewer #1: Yes

Reviewer #2: Yes

3. Has the statistical analysis been performed appropriately and rigorously? 

Reviewer #1: N/A

Reviewer #2: N/A

4. Have the authors made all data underlying the findings in their manuscript fully available?

Reviewer #1: Yes

Reviewer #2: Yes

5. Is the manuscript presented in an intelligible fashion and written in standard English?

Reviewer #1: Yes

Reviewer #2: Yes

6. Review Comments to the Author

Reviewer #1: The authors have answered all my comments and I have no further issue.

Just one very minor change to caption of Figure 1, the following suggestion reads better,

"We will show below that the concentration of species E, in control engineering terms called the manipulated variable (generally assigned the variable U in control community), is proportional to the integrated error..."

Reviewer #2: Thank you for addressing all the review points. Although I would like to recommend publication of this paper, I still have a couple of questions. (Please see the attached file) If the authors can address these questions in their final version, it would be great for readers who are particularly interested in mathematical analysis of the proposed controllers.

7. PLOS authors have the option to publish the peer review history of their article (what does this mean?). If published, this will include your full peer review and any attached files.

Reviewer #1: No

Reviewer #2: No

<gdiv></gdiv>

---

## [Editor Report · Acceptance letter]

26 Feb 2021

PONE-D-20-32544R1 

An amplified derepression controller with multisite inhibition and positive feedback 

Dear Dr. Ruoff:

I'm pleased to inform you that your manuscript has been deemed suitable for publication in PLOS ONE. Congratulations! Your manuscript is now with our production department. 

Kind regards, 

on behalf of

Dr. Jae Kyoung Kim 

Academic Editor

PLOS ONE